# SARS-CoV-2 hijacks a cell damage response, which induces transcription of a more efficient Spike S-acyltransferase

Francisco S. Mesquita [1,3] ✉, Laurence Abrami [1,3], Lucie Bracq[1], Nattawadee Panyain [1], Vincent Mercier [1,2], Béatrice Kunz[1], Audrey Chuat[1], Joana Carlevaro-Fita[1], Didier Trono [1] & F. Gisou van der Goot [1] ✉

SARS-CoV-2 infection requires Spike protein-mediated fusion between the viral and cellular membranes. The fusogenic activity of Spike depends on its post-translational lipid modification by host S-acyltransferases, predominantly ZDHHC20. Previous observations indicate that SARS-CoV-2 infection augments the S-acylation of Spike when compared to mere Spike transfection. Here, we find that SARS-CoV-2 infection triggers a change in the transcriptional start site of the *zdhhc20* gene, both in cells and in an in vivo infection model, resulting in a 67-amino–acid-long N-terminally extended protein with approx. 40 times higher Spike acylating activity, resulting in enhanced fusion of viruses with host cells. Furthermore, we observed the same induced transcriptional change in response to other challenges, such as chemically induced colitis and pore-forming toxins, indicating that SARS-CoV-2 hijacks an existing cell damage response pathway to optimize it fusion glycoprotein.

Infection by ß-coronaviruses (β-CoVs), such as the causative agent of the COVID-19 pandemic, SARS-CoV-2, have been extensively studied, yet the dependence on host factors remains incompletely understood. These large, enveloped RNA viruses enter host cells through interactions between the trimeric envelop glycoprotein Spike and host receptors such as ACE-2, followed by Spike-mediated fusion of the viral and cellular membranes. Like most other envelop glycoproteins[1], Spike undergoes multiple post-translational modifications, in particular S-acylation, which attaches a medium chain fatty acid via a thioester bond to cysteine residues in its cytoplasmic tail. This short tail contains no less than 10 such residues within a 20-amino acid stretch[2], full acylation of which would result in 30 fatty acids per Spike trimer. Acylation has been shown to influence the lipid composition of the virions and to drastically enhance the fusogenic capacity of the produced viruses[3–6]. The efficiency of Spike S-acylation is therefore determinant for infectivity.

During our previous work, we observed that Spike S-acylation is higher in cells infected with SARS-CoV-2 when compared to that of Spike transfected into the same cells[4]. We and others have shown that

S-acylation of Spike is mediated predominantly by the ZDHHC20 acyltransferase, with a contribution from ZDHHC9[4,6,7]. Here we show that SARS-CoV-2 infection, whether in human cells or in mice, triggers the use of an upstream transcriptional start site of the *zdhhc20* gene, leading to the production of a N-terminally extended enzyme, ZDHHC20[Long]. The expression of ZDHHC20[Long] is also triggered following chemically induced colitis and pore-forming toxins, suggesting that the transcriptional regulation of ZDHHC20 is part of a "damage" response pathway. In the context of SARS-CoV-2 infection, expression of ZDHHC20[Long] drastically enhances Spike acylation, and thus fatty acid decoration of the Spike trimers, which in turn augments their fusogenic capacity.

## Results

### Comparison of Spike S-acylation during infection vs. transfection

In our previous study aimed at characterizing Spike S-acylation, we had hints that the degree of S-acylation of the 10 available sites on SARS-CoV-2 Spike differed between infection and transfection[4]. This is

[1]Global Health Institute, School of Life Sciences, EPFL, Lausanne, Switzerland. [2]ACCESS, Department of Biochemistry, University of Geneva, Geneva, Switzerland. [3]These authors contributed equally: Francisco S. Mesquita, Laurence Abrami. ✉e-mail: francisco.mesquita@epfl.ch; gisou.vandergoot@epfl.ch

evident when using PEGylation, an assay that indicates the relative abundance of acylated proteins within the population, by replacing acyl chains with 5-kDa PEG[8] (after hydroxylamine treatment, Methods section), which produces molecular weight shifts corresponding to the number of acylated cysteines per Spike molecule. Visualized by western blot, we found that Spike undergoes extensive S-acylation in infected cells, with a complete band shift of the Spike S2 band denoting that all Spike molecules are massively acylated following their synthesis, and that the attachment of up to ten PEG molecules may hinder migration of fully modified full-length protein in these gels (Fig. 1a)[4]. In contrast, when a Spike-expressing vector was transfected into cells, Spike S-acylation is readily detectable by [3]H-palmitate incorporation[4], yet we could not detect major band shifts upon PEGylation (Fig. 1a). This indicates that S-acylation of Spike population is much less pronounced following transfection than infection.

## SARS-CoV-2 triggers a change in the transcriptional start site of *zdhhc20*

To understand how the efficiency of Spike S-acylation increases in infected versus transfected cells, we monitored the expression of the ZDHHC20 acyltransferase by western blot. In infected Vero E6 (Green monkey) cells, a cell line classically used for SARS-CoV-2 infection assays, the enzyme migrated as multiple isoforms in addition to the 42 kDa protein also detected in control cells, with an abundant >45 kDa band and fainter ~55 kDa bands (Fig. 1b). Such higher molecular weight species of ZDHHC20 became visible at 8 h post infection of Vero E6 cells (Fig. 1c, d) and could also be seen upon infection of more physiologically relevant cells such as human lung-derived Calu-3 (Supplementary Fig. 1a) or primary airway human epithelial cells (Fig. 1e). The appearance of higher molecular weight ZDHHC20 bands preceded significant detection of Spike or SARS nucleocapsid protein (N, Fig. 1e). This was not specific to the SARS-CoV-2 B.1-strain, since both Delta-Δ (B1.617.2) and Omicron-O (BA.1) triggered this same effect in ACE-2/TMPRSS2 expressing HEK293T cells (Supplementary Fig. 1b).

The green monkey ZDHHC20 protein sequence reported in UniProt (A0A0D9RZN5) carries an N-terminal extension that is not reported for the human protein (Q5W0Z9) (Supplementary Fig. 1c). We therefore analyzed the human *zdhhc20* gene and found three additional in-frame ATG sequences in the 5′ region, predicted to lead to proteins of 49, 62, and 64 kDa in addition to the 42 kDa canonical species (Fig. 1f). Since these additional putative translational start sites were all located upstream of the 5′ end of the annotated transcript, we performed 5′ Rapid Amplification of cDNA Ends (RACE) in infected cells for 5′-extended *zdhhc20* transcripts (see Methods). Some >15% of the RACE products included all 3 ATGs, spanning >6500 bp upstream of the annotated start site (Fig. 1f, Supplementary Fig. 1d, and Supplementary Table S2). We next quantified *zdhhc20* mRNA by qPCR. While primers to the coding sequence led to equivalent *zdhhc20* mRNA levels in control and infected cells, primers to the 5′ extensions showed enhanced expression during infection (Fig. 1f, g and Supplementary Fig. 1d, e).

To confirm that infected cells express a protein with an in-frame N-terminal extension, we generated cDNA constructs of the most abundant, 49 kDa, long-form (ZDHHC20[Long]) with an N-terminal myc-tag as well as a polyclonal antibody to peptides present in the N-terminal human extension (Supplementary Fig. 1c, see "Methods"). The antibody recognized ZDHHC20[Long] expressed upon transfection of the tagged version of the enzyme, but not the short form, and given the sequence identity, the extended monkey ZDHHC20[Long] in infected Vero E6 cells (Fig. 1h, i). The ZDHHC20[Long] bands revealed by the Long-specific and the polyclonal anti-ZDHHC20 antibodies (all) migrated similarly to the >45 kDa band of the ectopically expressed myc-ZDHHC20[Long] (Fig. 1h and Supplementary Fig 1f). Thus, SARS-CoV-2 infection promotes expression of *zdhhc20* transcripts with longer 5′ regions, coding for a ZDHHC20 protein harboring N-terminal

extensions, of which ZDHHC20[Long], with a 67-amino acid extension, is the most abundant.

To understand this change in transcriptional start site (TSS), we analyzed chromatin state segmentation tracks from UCSC Genome Browser[9], which integrates ChIP-seq data across nine human cell types (see data availability section). This showed that promoter activity was almost exclusively predicted in the region neighboring the human annotated start site, as depicted for embryonic stem cells (H1-hESC) (Fig. 1f and Supplementary Fig. 1g). However, in hepato-carcinoma HepG2 cells, additional promoter activity was reported ~6500 bps upstream from the annotated start site, proximal to the 5′ ends determined by 5′ RACE (Fig. 1f and Supplementary Fig. 1g). We probed HepG2 cells for ZDHHC20 protein expression and found constitutive expression of ZDHHC20[Long] (Fig. 1j). To identify transcription factors (TF) binding to this upstream promotor region, we analyzed ChIP-seq Peak data for HepG2 cells in ENCODE (Supplementary Fig. 1g). We tested the potential involvement of a panel of these TFs and found that the expression of ZDHHC20[Long] in HepG2 cells was inhibited by siRNAs against FOXA1/2 or SP1 (Supplementary Fig. 1h) and by their pharmacologic inhibition both in HepG2 and during SARS-CoV-2 infection in Vero E6 cells (Fig. 1j, k). FOXA1 and SP1 were also upregulated during infection in both Vero E6 and Calu-3 cells (Supplementary Fig. 1i), but their overexpression was not sufficient to trigger ZDHHC20[Long] production in HeLa cells (Supplementary Fig. 1j).

Altogether, these results demonstrate that SARS-CoV-2 infection triggers the FOXA1- and SP1-mediated activation of an upstream promoter of the *zdhhc20* gene, resulting in the production of a 67-amino-acid N-terminally extended acyltransferase product. This is probably not restricted to the species studied here, human, monkey and mouse, since genomic analyses indicated that synthesis of N-terminally extended versions of ZDHHC20 might be conserved in a wide range of animal species, including many other mammals, fishes, amphibians, and the fruit fly (Fig. 1l).

## SARS-CoV-2 infection triggers the production of ZDHHC20[Long] in mice

Before probing for the expression of ZDHHC20[Long] in a mouse model of SARS-CoV-2 infection, we analyzed the expression of murine ZDHHC20 under physiological conditions. All tissues tested expressed exclusively ZDHHC20[Short], except for the small intestine (duodenum and ileum), where ZDHHC20[Long] was also detected (Fig. 2a). We could not probe these tissue samples with the antibody we generated towards the human extension of ZDHHC20 because it does not cross-react with mouse ZDHHC20[Long] due to sequence differences (Supplementary Fig. 2a). Consistent with the expression of ZDHHC20[Long], FOXA1 expression was also higher in the duodenum compared to kidney or colon (Supplementary Fig. 2b). Publicly available murine *zdhhc*20 mRNA sequences[9] confirmed the existence of transcripts with long 5′UTRs covering alternative start sites (Supplementary Fig. 2c). To evaluate the distribution of *zdhhc20*[Long] mRNAs in the duodenum, we used RNAscope probes against the extended 5′ region (Supplementary Fig. 2c). We confirmed the expression of a longer transcript of *zdhhc20* in the duodenum (Fig. 2b), with a higher level in the crypt region and at the base of the villi (Supplementary Fig. 2d). Analysis of commercially available RNA from a panel of human tissues also indicated that *zdhhc20* transcripts with longer 5′ regions were significantly increased in the human small intestine (Fig. 2c). Thus, both in mouse and human, most tissues express ZDHHC20[Short], with the exception of the small intestine, which expresses ZDHHC20[Long].

We next infected human-ACE-2 transgenic mice intranasally with SARS-CoV-2[10]. We analyzed the *zdhhc20* mRNA in lungs using primers complementary to the coding region or the 5′UTR (Supplementary Fig. 2c). Increased expression of 5′-extended *zdhhc20* transcripts was visible already 1-day post infection, with a peak at 3/4 days (Fig. 2d). An increase of extended *zdhhc20* mRNA transcripts could be detected in

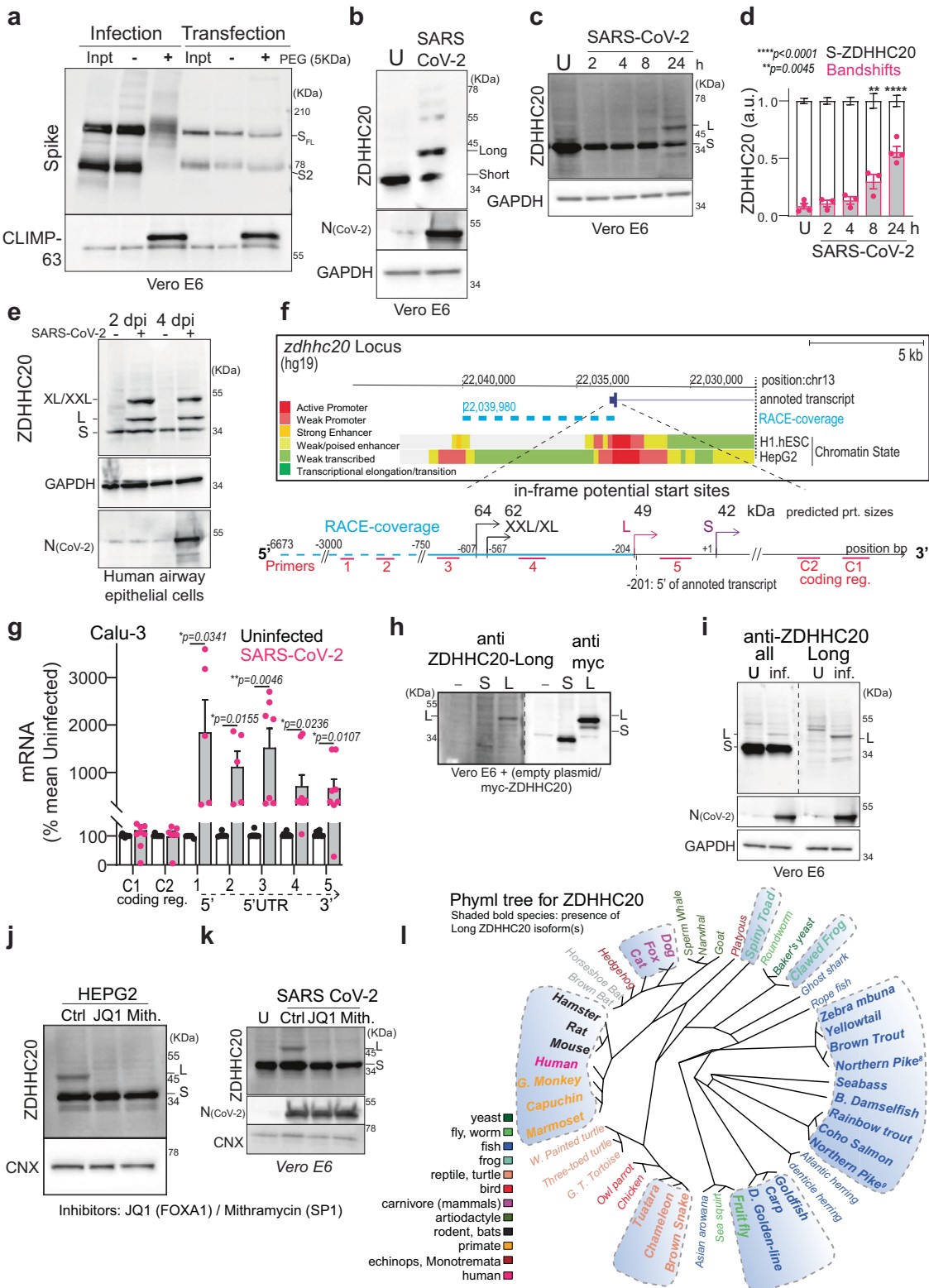

all tested tissues of infected mice, with the exception of the brain and cerebellum (Fig. 2d and Supplementary Fig. 2e), but the kinetics of appearance varied between organs as illustrated for the spleen (Fig. 2d). Expression of ZDHHC20[Long] protein was observed in all tested tissues by western blot analysis, with the exception of the kidney (Fig. 2e). We also probed for the viral nucleocapsid N protein, which was detected in lung, brain, cerebellum and colon, as reported[18,19], but undetectable in the kidney, spleen, liver and small intestine. Despite low levels of viral genomes, marked pathological

features have been reported in these organs, in both humans and the human-ACE-2 transgenic mice[11–16]. Consistently, we also observed that viral loads were highly variable between tissues, and throughout time, with high viral copy numbers in the lung, moderate levels in the brain and low levels in the spleen (Supplementary Fig. 2f). Jointly, these observations show that under physiological conditions, ZDHHC20[Long] is only expressed in specific mouse tissues, such as the small intestine, but becomes expressed in most tissues/organs in SARS-CoV-2-infected mice. At this stage, we cannot exclude that

**Fig. 1 | SARS-CoV-2 triggers a change in the transcriptional start site of the _zdhhc20_ gene. a** Acyl-PEG exchange, Vero E6, infected MOI = 0.1, 24 h or Spike-transfected. Western blot (WB) of cell extract (Inpt), PEG-tagged protein ( + PEG), and control (−) after hydroxylamine treatment. Spike (full length -SFL and cleaved S2) or control Climp-63. **b** WB of ZDHHC20, nucleocapsid (N) and GAPDH (control) of Vero E6, uninfected (U) or infected as in (**a**). Short (42 KDa, S) and Long (49 KDa, L) ZDHHC20 are indicated. **c** Same as (**b**) through time. **d** Quantification of ZDHHC20 band shifts. Results are mean ± SEM, and each dot represents one independent experiment of $n = 4$ (U and 24 h) and $n = 3$ (2, 4, 8 h). P values comparing to Uninfected were obtained by two-way ANOVA, Dunnet's multiple comparison. **e** Same as in (**b**) in human airway epithelial, days post infection (dpi) (XL/XXL-62/69 KDas). **f** _zdhhc20_ locus (version hg19) indicating in-frame transcription start sites (TSS), coding for XXL/XL, Long and Short; 5′-RACE coverage (blue); Transcript from GENCODE Genes track (V4 0lift37); Chromatin State segmentation from ENCODE for H1.hESC and HepG2. **g** mRNA quantification probing different locations in _zdhhc20_ transcripts (coding region: C1, C2; 5′ UTR: 1−5 cover increasing lengths of 5′ ends (Supplementary Fig. 1D) in Calu-3 uninfected or infected as in (**a**). Results are mean ± SEM, and each dot represents one independent experiment. P obtained by unpaired, two-tailed, t tests, (C1, $n = 10$; C2 and primer 3, $n = 7$; primers 1 and 2, $n = 5$; primers 4 and 5, $n = 8$); **h** WB of ZDHHC20 on Vero E6 transfected with myc-ZDHHC20 for 24 h, probed with anti-ZDHHC20$^{Long}$ (left) or anti-myc antibodies (right) **i** WB as in (**b**) (left) compared to ZDHHC20$^{Long}$-specific WB (right). **j** WB of ZDHHC20 and Calnexin (control, CNX) on HEPG2 treated 24 h with 100 nM JQ1 (FOXA1 inhibitor) or 100 nM mithramycin (SP1 inhibitor). **k** WB in infected Vero E6 as in (**j**). **l** Phyml tree for ZDHHC20 with the presence of ZDHHC20$^{Long}$ (shaded blue). Common names indicated (see also Supplementary Table S1). All source data are provided as a Source Data file or Supplementary information.

paracrine signaling plays a role in ZDHHC20$^{Long}$ expression in tissue with low or no viral burden.

## ZDHHC20$^{Long}$ is expressed in mouse colon following chemically induced colitis

Given the proposed association between alternative transcription and stress[17], and the fact that ZDHHC20$^{Long}$ is expressed in most tissue/organs in SARS-CoV-2-infected mice, in particular the colon, we wondered whether the expression of ZDHHC20$^{Long}$ could be triggered in this organ in response to another type of stress. For this we chose a well-established chemically induced colitis model[18] in which mice are treated orally with 3% dextran sodium sulfate (DSS) in the drinking water for 7 days. As described in the literature[18], the treatment led to diarrhea, rectal bleeding and body weight loss, characteristic of acute colitis (Supplementary Fig. 3a–d). The mice were subsequently given clear water. Consistent again with the literature, this led to the immediate stop of rectal bleeding and body weight loss and gradual recovery of the intestinal epithelium as assessed histologically by the presence of regenerating hypertrophic crypts labeled with the differentiated-colonocyte marker keratin-20 (Fig. 3a and Supplementary Fig. 3a–d). We assessed ZDHHC20 expression before and after the 10-day colitis experiment. We detected ZDHHC20$^{Long}$ and XL forms in the colon of DSS-treated mice by western blot (Fig. 3b). RNAscope analysis showed that the average mice levels of _zdhhc20_ transcripts with long 5′UTRs were similar between control and DSS-treated mice (Fig. 3a and Supplementary Fig. 3e). However, a closer examination of the different regions showed that _zdhhc20_ transcripts with long 5′UTRs were barely detected in highly damaged regions of DSS-treated mice, but increased locally in regions of the colon undergoing recovery, when compared to control untreated mice (Fig. 3a,c). This observation suggests that ZDHHC20$^{Long}$ expression might occur during the recovery phase.

The finding that both SARS-CoV-2 infection and colitis induce the expression of ZDHHC20$^{Long}$ raises the possibility that this transcriptional regulation is part of a general response to danger/damage. To test this hypothesis, we choose one more challenge that we have extensively studied in the past, the treatment of cells with a pore-forming toxin[19]. Pore-forming toxins are the largest class of bacterial protein toxins[19]. Each member forms well-defined pores in the plasma membranes of their specific target cells, eliciting specific responses that promote cell survival and recovery of plasma membrane integrity, while overall promoting the bacterial infection[20,21]. We choose aerolysin, produced by _Aeromonas hydrophila_, a bacterium that can lead to severe diarrhea in children[22]. While it makes small pores in membranes, these are extremely stable structures, and thus cells will die after 3 to 4 h of continuous exposure, but will survive if exposed to a transient pulse[23]. Upon continuous exposure of Vero E6 cells, and before extensive cell death, we could not detect any change in ZDHHC20 expression (Supplementary Fig. 3f). However, upon transient exposure to aerolysin, which allows recovery of plasma membrane integrity, we observed a conversion of ZDHHC20 from the Short to the Long form both in Vero E6 (Fig. 3d) and HeLa cells (Supplementary Fig. 3g). Expression of ZDHHC20$^{Long}$ was visible within 5 h, peaked around 24 h and disappeared after 48 h (Supplementary Fig. 3h). As observed for the DSS-induced colitis, the expression of ZDHHC20$^{Long}$ appears more pronounced in the recovery phase, than the damage phase.

The fact that the switch from the Short to the Long form of ZDHHC20 can be triggered by three very different challenges, viral infection, chemical-induced colitis and pore formation by aerolysin, suggest that this transcriptional response is part of a rather general damage repair pathway, which remains to be explored.

## ZDHHC20$^{Long}$ is longer lived and retained in the ER

Confident that ZDHHC20$^{Long}$ is a physiologically relevant isoform of the enzyme, based on the above observations, we next focused on understanding the consequences of the N-terminal 67-amino-acid extension, first on the enzyme itself and subsequently in the context of Spike.

We first performed metabolic labeling of ZDHHC20 with $^{35}$S Cys/Met. Transfection of cells with equivalent amounts of the corresponding plasmid DNAs of Long or Short ZDHHC20 led to the synthesis of the same amounts of protein (Supplementary Fig. 4a). However, total ZDHHC20$^{Long}$ levels were approximately five times higher than ZDHHC20$^{Short}$ (Fig. 4a), which could be explained by a twofold increase in the apparent protein half-life, determined by $^{35}$S Cys/Met metabolic pulse-chase analysis (Fig. 4b).

We next analyzed the subcellular localization of ZDHHC20$^{Long}$, knowing that the canonical human ZDHHC20 (ZDHHC20$^{Short}$) accumulates in the Golgi, in vesicles, and at the plasma membrane[4,24] (Fig. 4c, d and Supplementary Fig. 4b). Surprisingly, ZDHHC20$^{Long}$ localized exclusively to the Endoplasmic Reticulum (ER) (Fig. 4c, d and Supplementary Fig. 4b–d). To determine which part of the N-terminal extension was responsible for this drastic change in localization, we generated reporter constructs encoding chimeric proteins with the Short ZDHHC20 cytosolic tail, the first transmembrane domain of the protein fused to a GFP variant harboring a N-glycosylation site on the luminal side (Fig. 4e). On the cytosolic side, we added or not full-length or truncated versions of the N-terminal extension of ZDHHC20$^{Long}$ (Fig. 4e). Short and Long reporters behaved in accordance to the full-length ZDHHC20 proteins, but with a more exclusive Golgi distribution for the Short reporter (Fig. 4f and Supplementary Fig. 4e). The Δ39−67 deletion mutant, harboring the N-terminal half of the 67-amino-acid extension also localized entirely to the Golgi, while the Δ1−38 and Δ1−53 deletion carrying C-terminal parts of the extension localized to the ER (Fig. 4g, h). We further truncated the Δ1−53 mutant by 4 residues, namely PERW (Δ1−57), and this abolished ER localization. Mutation of these 4 residues to alanine in the Long reporter was sufficient to shift its localization from ER to Golgi (Fig. 4h, i). Quantitative high-throughput microscopy showed that a full-length AAAA

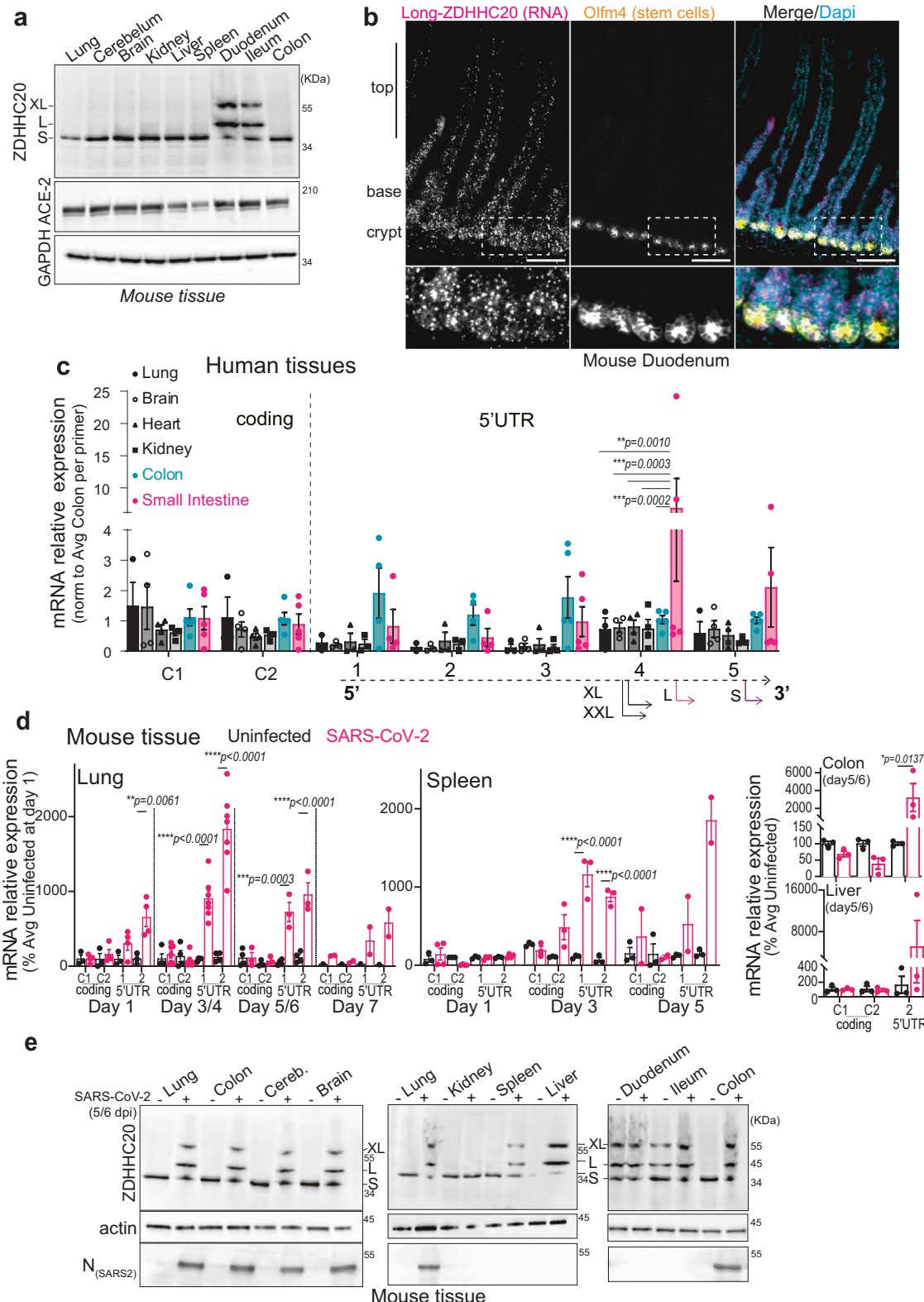

ZDHHC20$^{Long}$ mutant similarly shifted its localization from ER to Golgi, confirming the requirement for the PERW motif (Fig. 4j).

To test whether the N-terminal extension leads to ER retention or ER retrieval, we used the RUSH system, wherein ZDHHC20 reporters were hooked to the ER via a streptavidin-binding peptide (SBP) and the co-expression of a streptavidin-tagged ER "hook" protein (see "Methods")[25]. Biotin was added to release the ZDHHC20 reporters

from the ER hook and cells were monitored by live microscopy to determine whether transport to the Golgi could be observed. The ZDHHC20$^{Short}$-RUSH reporter was transported from the ER to the Golgi (Fig. 4k bottom left quantification panel, Supplementary Fig. 4f, and Supplementary Movie 1). In contrast, we could not detect any transport of the ZDHHC20$^{Long}$-RUSH reporter to the Golgi, it remained in the ER even after 30 min (Fig. 4k and Supplementary Movie 2).

**Fig. 2 | SARS-CoV-2 infection triggers expression of ZDHHC20[Long] in mice. a** WB of ZDHHC20, ACE-2, and GAPDH loading control in mouse tissue extracts. Short (44 KDa, S), Long (50 KDa, L), and XL (53 KDa) ZDHHC20 forms are indicated. **b** Longitudinal sections of mouse duodenum are labeled with an RNAscope probes for 5′ UTR *zdhhc20* RNA, DAPI, and antibodies against Olfm4 (marking intestinal stem cells). Scale bars 100 µm. **c** mRNA quantification in human tissues, using primers probing for different locations in *zdhhc20* transcripts (coding region: C1, C2; 5′ UTR: 1–5 cover increasing lengths of 5′ ends see also Supplementary Fig. 1d). Results are mean ± SEM and each dot represents one independent pool of human-derived tissue-specific mRNA. Lung, n = 3, Brain and Heart, n = 4, and Colon and Small Intestine, n = 5. *P* values comparing primer-specific expression between tissues were obtained by two-way ANOVA with Tukey's multiple comparison. **d** mRNA quantification in different murine tissues uninfected or infected intranasally with $10^3$-$10^4$ (plaque-forming units−PFU) of SARS-CoV-2. Tissues were harvested at indicated times post infection and the extracted mRNA analyzed using primers for different locations in *zdhhc20* transcripts (coding region (C1, C2) or the 5′ UTR [1–2] see also Supplementary Fig. 2c. **e** WB analysis of tissue extracts from control uninfected mice or mice infected as in (**d**). Tissues were harvested at 5−6 days post infection. Actin was used as loading control.). Results are mean ± SEM and each dot represent one of 4 independent mice. *P* values were obtained by two-way ANOVA with Sidak's multiple comparison. For all *P < 0.05, **P < 0.01, ***P < 0.001, and ****P < 0.0001, and source data are provided as a Source Data file or Supplementary information.

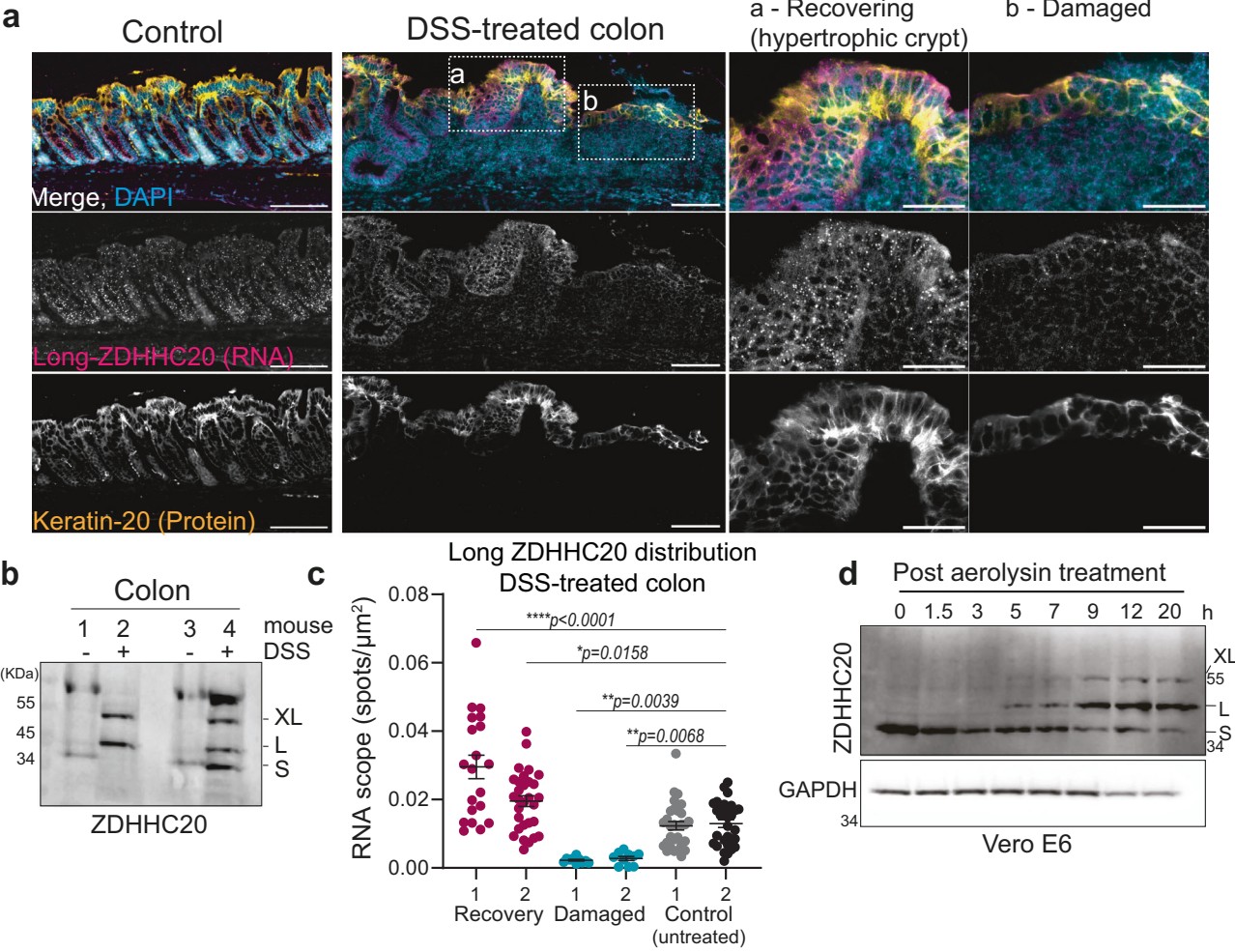

**Fig. 3 | ZDHHC20[Long] expression is triggered in mouse colon following chemically induced colitis. a** RNAscope analysis of DSS-treated mice using probes for 5′ UTR *zdhhc20* RNA, DAPI staining, and anti-Keratin-20 antibodies (to mark intestinal epithelial cells). Scale bars 100 µm. **b** WB analysis of mice treated or not with DSS for 7 days followed by 3 days recovery, showing ZDHHC20 expression from colons from two independent mice for each condition S, L, and XL forms are indicated. **c** Representative quantification (two of three independent mice) of Long-*zdhhc20* RNAscope spots per µm² in specific regions from colon sections from mice untreated or DSS-treated as in (**a**). Results are mean ± SEM and each dot corresponds to one field-of-view selected from recovering (n = 20 or 30), damaged (n = 10) or control untreated (n = 30) zones represented in (**b**). *P* values comparing with Control-2 were obtained by one-way ANOVA with Dunnet's multiple comparison. **d** WB analysis of ZDHHC20 expression (S, L and XL forms are indicated) in Vero E6 cells treated 1 h with 10 ng/ml of proaerolysin at 37 °C, washed, and further incubated at 37 °C for the indicated time. For all *P < 0.05, **P < 0.01, ***P < 0.001, and ****P < 0.0001, and source data are provided as a Source Data file or in Supplementary information.

These morphological observations indicate that the ZDHHC20 N-terminal extension retains the protein in the ER, as opposed to retrieving it from the Golgi following ER exit. To confirm that the ZDHHC20 reporters that accumulate in the ER never reached the Golgi, we probed them biochemically for their sensitivity to Endoglycosidase-H (EndoH), which cleaves simple N-linked sugars generated in the ER but not complex sugars resulting from modifications in the Golgi[26]. All reporters migrated as multiple bands, for reasons that remain unclear. Importantly, the Short reporter as well as Δ39−67 and Δ1−57 deletion mutants and the Long-AAAA mutant, in part migrated as a higher molecular weight smear, characteristic of complex Golgi-generated sugars, which were insensitive to EndoH

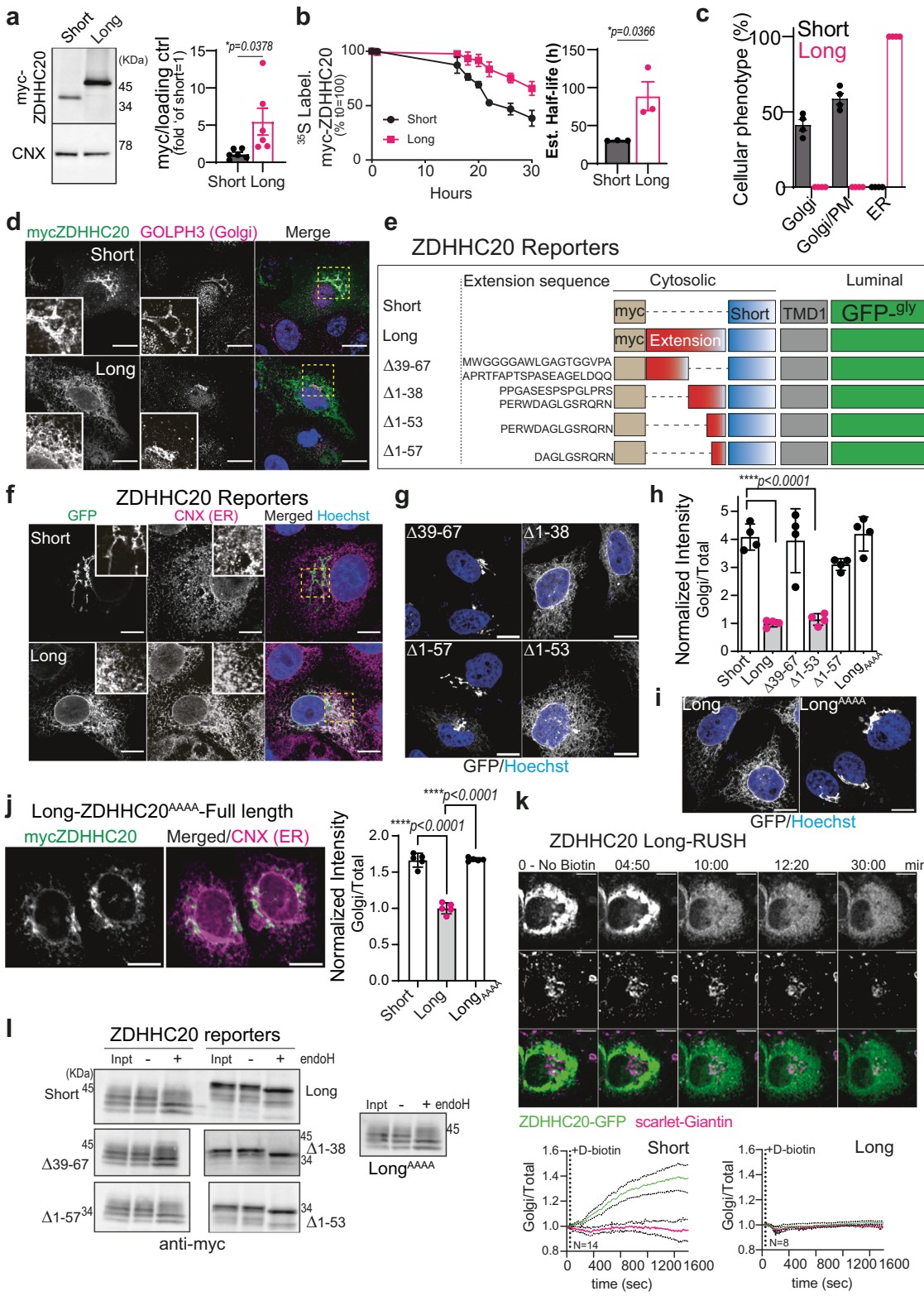

treatment (Fig. 4l). In contrast, the Long reporter, and the Δ1–38 and Δ1–53 deletion mutants did not show the higher molecular weight smear, and all protein bands were sensitive to EndoH treatment (Fig. 4l), independently confirming the RUSH experiments that these forms had not reached the Golgi.

Altogether these observations show that ZDHHC20[Long] accumulates in the ER via a specific retention motif, and has an extended half-life that leads to higher protein levels compared to ZDHHC20[Short].

## ZDHHC20[Long] has greatly enhanced performance in modifying Spike

Having established that the N-terminal extension present in ZDHHC20[Long] leads to higher proteins levels than ZDHHC20[Short] and to ER localization, we assessed its ability to S-acylate Spike, by monitoring incorporation of [3]H-palmitate. Palmitate incorporation was observed with both forms of the enzyme. However, in cells expressing ZDHHC20[Long], [3]H-palmitate incorporation into Spike was much faster

**Fig. 4 | The N-terminal extension of ZDHHC20[Long] controls its abundance and localization. a** WB of myc-ZDHHC20 and calnexin (CNX) on Vero E6 expressing ZDHHC20[Short] (Short) or ZDHHC20[Long] (Long). myc-ZDHHC20/loading control ratio are mean ± SEM, and each dot represents one of $n = 6$ independent experiments. $P$ values obtained by unpaired two-tailed $t$ test. **b** 35S-Met/Cys apparent decay in ZDHHC20 immunoprecipitation fractions from HeLa cells expressing myc-ZDHHC20 (Short or Long), normalized to $T = 0$, corresponds to mean ± SD, of $n = 3$ independent experiments. Estimated half-lives extracted from the individual experiments using non-linear regression with one-phase decay. Results are mean ± SEM and $P$ values obtained by unpaired two-tailed $t$ test. **c** Qualitative quantification of myc-ZDHHC20 (Short or Long) distribution (plasma membrane-PM) in Vero E6. Results are mean ± SEM, of $n = 4$ independent experiments where a total of 1235 (Short) and 814 (Long) cells were counted. **d** Immunofluorescence (IF) of Vero E6 expressing myc-ZDHHC20 (Short or Long) labeled for myc (ZDHHC20) and Golgi marker GOLPH3, scale bar: 10 μm. **e** Illustration of ZDHHC20 reporters with N-terminal myc-tag, the full 67-amino acid ZDHHC20 N-terminal extension (Long), or partial deletions thereof, or short N-terminal tail; and first transmembrane (TM) domain, and luminal glycosylated GFP. **f** IF of Vero E6 expressing ZDHHC20 reporters labeled for ER marker CNX and nuclear-stained with Hoechst, scale bar: 10 μm. **g** IF of HeLa cells expressing the ZDHHC20 reporters illustrated in E, scale bar: 10 μm. **h** high-throughput quantification microscopy of HeLa cells as depicted in (**g**). Data represent ratio of GFP signal between Golgi and cytoplasm

normalized by to ZDHHC20[Long]-reporter. Data, representative of two independent experiments, were obtained from 4 wells with 9 images per well. Each dot represents the mean from one independent well, results are mean ± SEM and $P$ values obtained by one-way ANOVA with Tukey's multiple comparison. Total number of cells analyzed: S: 3755, L: 3096, Δ39–67: 2990, Δ1–38: 3213, Δ1–53: 3078, Δ1–57: 3681. **i** IF of HeLa cells expressing ZDHHC20[Long]-reporter with PERW/AAAA mutation, scale bar: 10 μm. **j** IF of HeLa cells expressing full-length ZDHHC20[Long] with PERW/AAAA mutation (Long[AAAA]) labeled for myc and CNX, scale bar: 10 μm. High-throughput quantification in HeLa cells as depicted in (**d, j**). Data were obtained from five wells with 25 images per well. Each dot represents the mean from one independent well, results are mean ± SEM and $P$ values obtained by one-way ANOVA with Tukey's multiple comparison, results are mean ± SEM. Total number of cells analyzed: S:19478, L:2041 and Long[AAAA]: 19166. **k** Time-lapse microscopy images of Vero E6 co-expressing (24 h) the Golgi marker Scarlet-Giantin, and ZDHHC20[Long]-RUSH-GFP reporter with ER hook. Synchronized trafficking was monitored upon D-biotin addition ($T_0$). Normalized GFP ratio at the Golgi (Scarlet-Giantin) throughout time ($T_0 = 1$; every 10 s for 30 min) for $n = 14$ ZDHHC20[Short]- or $n = 8$ ZDHHC20[long]- RUSH reporter expressing independent cells (see also Supplementary Fig. 4f and Supplementary Movies 1 and 2). **l** EndoH assay: WB on Vero E6 cell expressing ZDHHC20 reporters. Cell extracts (40 μg) were treated (+) or not (−) with EndoH. For all \*$P < 0.05$, \*\*$P < 0.01$, \*\*\*$P < 0.001$, and \*\*\*\*$P < 0.0001$, source data are provided as a Source Data file or in Supplementary information.

and also plateaued at a level ≈37 times higher than in cells expressing ZDHHC20[Short] (Fig. 5a). This shows that in the presence of ZDHHC20[Long], the Spike population, as a whole, acquires palmitate on more cysteines. Our previous mutagenesis analysis indicates that when acylation starts on a given Spike molecule, it tends to proceed to near completion on the ten cysteines of the cytosolic domain (CCXXXCCXCXXXCCXCXXCC)[4]. This is consistent with our previous work showing that the presence of successive cysteines leads to cooperative S-acylation[27], presumably because when cysteines are sufficiently close, acyl transferases can modify more than one during an enzyme-substrate contact event. The increased 3H-palmitate incorporation we observed in the presence of ZDHHC20[Long], therefore suggests that more individual Spike molecules undergo acylation, on all ten cysteines. Our previous observations also indicate that acylation of successive cysteines protects proteins from deacylation[27]. This altogether would predict that the faster Spike gets acylated, such as in ZDHHC20[Long]-expressing cells, the more resistant it becomes to deacylation. To test this, we conducted pulse-chase experiments following 3H-palmitate labeling. In cells expressing ZDHHC20[Short], Spike lost ≈78% of the 3H-palmitate within 6 h (Fig. 5b), as observed previously[4]. In cells expressing ZDHHC20[Long], only ≈33% of the Spike-associated 3H-palmitate was lost (Fig. 5b). Spike deacylation is thus significantly reduced in cells expressing ZDHHC20[Long].

However, as the deacylation of Spike still occurs, to monitor the full acylation capacity, we performed 3H-palmitate labeling in the presence of the broad deacylation inhibitor Palmostatin B (PalmB). Comparison between the steady state 3H-palmitate incorporation values can provide information on the relative number of S-acylated cysteines within the Spike population, under different conditions. The plateau values of 3H-palmitate incorporation into the Spike population significantly increased both in cells expressing ZDHHC20[Short] and ZDHHC20[Long] in the presence of Palmostatin B (Fig. 5c, d). We assumed that the 3H-palmitate plateau value reached in Palmostatin-B-treated cells expressing ZDHHC20[Long] corresponded to 100% of the Spike molecules within the global population modified on all ten cysteines (Fig. 5e). With this assumption, Spike would be modified, on average over the population, on: (i) 5.7 cysteines in cells expressing ZDHHC20[Long] in the absence of Palmostatin B; (ii) 0.16 cysteines in cells expressing ZDHHC20[Short]; and (iii) 0.4 cysteines in the same cells treated with Palmostatin B (Fig. 5e).

These estimations reveal that ZDHHC20[Short] is extremely inefficient in acylating the Spike population, likely modifying only very few individual Spike molecules, albeit on all ten cysteines. We confirmed

this inefficiency using PEGylating of extracts from cells expressing exclusively ZDHHC20[Short]. No significant band shift of Spike was observed (Fig. 5f). Even when infecting double ZDHHC9/20 KO cells complemented with ZDHHC20[Short], PEGylation did not shift the Spike bands (Fig. 5g). In contrast, upon either transfection or infection of cells expressing ZDHHC20[Long], Spike underwent massive PEGylation leading to almost full disappearance of the full-length and S2 Spike bands (Fig. 5f, g).

Altogether, these observations show that ZDHHC20[Long] is a drastically more potent enzyme for S-acylation of Spike. This is not only due to the increased expression of ZDHHC20[Long], since when adjusting the amount of transfected plasmid DNA to reach similar protein levels (Supplementary Fig. 5a), ZDHHC20[Long] was still far more efficient in S-acylating Spike (Fig. 5a).

## ZDHHC20[Long] leads to more fusogenic SARS-CoV-2 viruses and viral-like particles

We next determined the consequence of the differential acylation of Spike on viral fusion capacity. To identify the specific contributions of ZDHHC20[Short] and ZDHHC20[Long], we expressed these enzymes in Vero E6 cells after KO of endogenous ZDHHC20 and ZDHHC9. Following infection with SARS-CoV-2, the supernatants containing viruses were collected at 48 h. The relative abundance of structural SARS-CoV-2 proteins, Spike, N and M was not significantly influenced by which acyltransferase was expressed (Supplementary Fig. 5b)[4]. Following normalization to their viral RNA content, these supernatants were used to infect Vero E6 cells. Viral RNA was measured 6 h post inoculation, corresponding to a single round of infection. Infection with virions produced by ZDHHC20[Long] expression cells induced an approximately twofold increase in viral RNA 6 h post inoculation, and thus in infectivity (Fig. 5h).

Since ZDHHC20 modifies other SARS-CoV-2 proteins[28] as well as antiviral proteins[29,30], we more specifically addressed the role of ZDHHC20-modified Spike using the viral-like particles (VLPs) established by the Gallagher lab[4,31,32]. These VLPs, based on a split reporter system engineered into the N protein of SARS-CoV-2, were produced from HEK293T ZDHHC20 KO cells silenced for ZDHHC9 expression and recomplemented with ZDHHC20[Short] or ZDHHC20[Long]. VLPs were normalized to their N-reporter content, and fusion dynamics were monitored by measuring bioluminescence as a function of time. VLPs displayed some fusogenic activity in the absence of Spike acyltransferases, which was moderately enhanced upon ZDHHC20[Short] expression (Fig. 5i) as observed previously using HIV-based viral

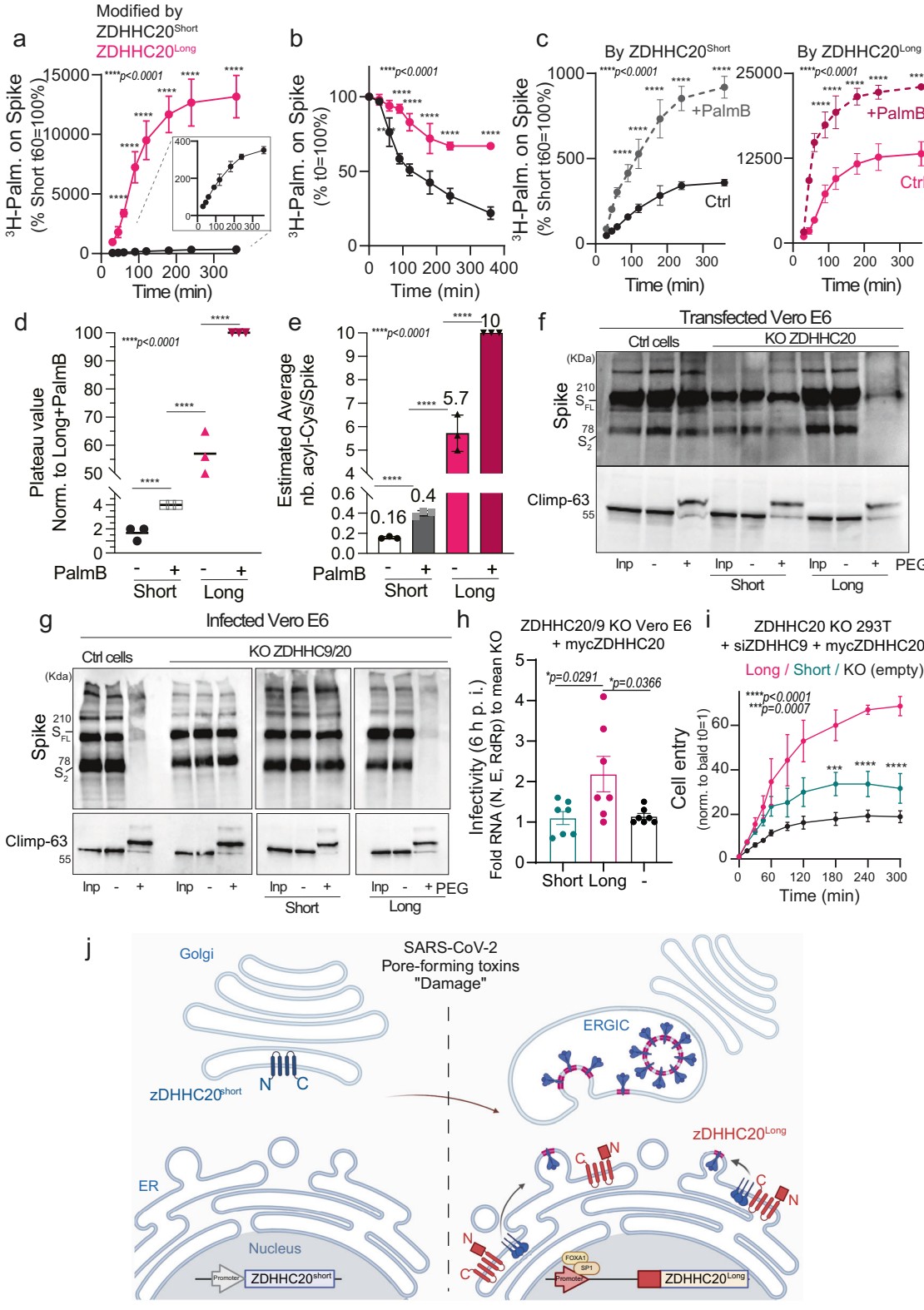

pseudotypes[4]. In contrast, a >threefold increase was observed when expressing ZDHHC20Long (Fig. 5h, i). We conclude that the ZDHHC20 enzyme isoform that is expressed at late stages of infection by SARS-CoV-2 ensures that Spike is extensively S-acylated to optimize its fusogenic capacity (Fig. 5j).

## Discussion

We here report that, following various challenges, a change occurs in the *zdhhc20* TSS, leading to longer transcripts carrying 1–3 additional in-frame translational start sites, enabling the expression of ZDHHC20 proteins with extended N-termini, the most abundant of which contains an additional 67 amino acids (illustrated in the model Fig. 5j). The potential for such N-terminally extended versions is conserved in different species. Alternative promoters are not unusual and have been found for about half of human genes[33,34]. The usage of alternative TSSs, recently described as a conserved stress-response mechanism, however, seems to be aimed at fine-tuning mRNA expression without altering the proteome[17], in contrast to what we observed here.

**Fig. 5 | Expression of ZDHHC20$^{Long}$ drastically increases Spike S-acylation and leads to more infectious viruses. a** Incorporation of $^3$H-palmitic acid in Spike-HA immunoprecipitation fractions normalized to $T = 60$ min from Vero E6, KO-ZDHHC20 recomplemented with ZDHHC20$^{Short}$ or ZDHHC20$^{Long}$. Inset−curve from ZDHHC20$^{short}$−cells. **b** $^3$H-palmitate turnover from Spike-HA from cells as in (**a**), labeled for 3 h pulse and chased as indicated. Values are set to 100% at $T = 0$. **c** Same, as in (**a**), in cells are pretreated or not (Ctrl) with Palmostatin B. For (**a**–**c**) results are mean ± SD of $n = 3$ independent experiments and $P$ values obtained by two-way ANOVA with Sidak's multiple comparison. **d, e** Average plateau values and correspondent estimated palmitoylated Cysteines derived from curves in (**c**). Data, normalized to 100%-modified or ten palmitoylated Cys for ZDHHC20$^{Long}$ cells treated with Palmostatin, are mean, **d** or mean +/− SEM, **e** of $n = 3$ independent experiments and $P$ values were obtained by one-way ANOVA with Tukey's multiple comparison. **f** Acyl-PEG exchange in Vero E6 control (Ctrl) or KOZDHHC20 recomplemented with ZDHHC20$^{Short}$ or ZDHHC20$^{Long}$ transfected with WT-Spike for 24 h. WB of Spike and control CLIMP-63 showing input extract (Inp), mass-tagged protein ( +PEG) and control (-mPEG) after hydroxylamine treatment (NH$_4$OH). **g** Same as in (**f**) in infected Vero E6 KO-ZDHHC9/20 recomplemented

with ZDHHC20$^{Short}$ or ZDHHC20$^{Long}$ (MOI = 0.1, 24 h). **h** Infectivity (viral RNA at 6 h p.i. of naive Vero E6) of virion supernatants from cells as in (**g**) adjusted to viral RNA content. Results normalized to infectivity of KO-ZDHHC20/9-derived supernatants are mean ± SEM, of $n = 7$ independent experiments. $P$ values were obtained by one-way ANOVA with Tukey's multiple comparison. **i** Representative viral-like particles-(VLP)-Cell entry assay of Suspensions of VLPs (produced in KO-ZDHHC20 HEK293T, siZDHHC9-depleted for 72 h−stably expressing ZDHHC20, Short/Long or empty plasmids-KO empty) adjusted by HiBit-N multiplicities, in ACE-2-LgBit-transfected HEK293T-ACE-2-TMPRSS2 target cells. Results are mean ± SEM of one representative experiment with three stock replicates repeated for three independent times. $P$ values were obtained by two-way ANOVA with Sidak's multiple comparison. **j** Model of the damage/repair (e.g., SARS-CoV-2)-induced transcriptional shift of ZDHHC20. The N-terminally extended ZDHHC20 isoform is expressed in a SP1- and FOXA1-dependent manner, at higher levels, in the ER and is more efficient in acylating Spike (BioRender.com full license). For all *$P < 0.05$, **$P < 0.01$, ***$P < 0.001$, and ****$P < 0.0001$, source data are provided as a Source Data file or in Supplementary information.

We found that expression of ZDHHC20$^{Long}$ depends of the SP1 and FOXA1 transcription factors. Interestingly, FOXA1 also coordinates the expression of the Spike receptor ACE-2 and the Spike priming factor TMPRSS2[35]. It remains unclear how FOXA1 and SP1 gain access to the promoter region located 6500 base pairs upstream from the annotated start site of the human *zdhhc20* gene. This could be due to specific virus-induced alterations of host genome architecture[36–39]. Alternatively, since the TSS of *zdhhc20* also changed in response to other cellular challenges, the modification in gene architecture might be part of a more general host response to stress, particularly during the repair/recovery phase.

The N-terminal extension of ZDHHC20$^{Long}$ lengthens the half-life of the protein, leading to fivefold increase in enzyme abundance. It also contains a 4-amino acid ER retention motif, PERW, that prevents transport to the Golgi and plasma membrane, where ZDHHC20 normally localizes. PERW-mediated retention could involve the binding to some ER-resident protein that remains to be identified.

In the specific context of SARS, these novel characteristics contribute to the highly enhanced capacity of ZDHHC20$^{Long}$ to modify Spike. ER localization likely makes ZDHHC20$^{Long}$ more adapted to Spike, as SARS-CoV-2 virus assembly occurs in the ER-Golgi intermediate compartment (ERGIC)[40,41] (Fig. 5j). Spike can reach the ERGIC either by anterograde trafficking from the ER or retrograde trafficking from the Golgi[42]. In cells expressing ZDHHC20$^{Short}$, located in the Golgi, Spike would require transport to the Golgi to become acylated, followed by subsequent retrograde trafficking to the ERGIC for incorporation into virions. In contrast, ZDHHC20$^{Long}$ expression enables Spike acylation in the ER shortly after protein synthesis, and subsequent anterograde delivery to the ERGIC for incorporation into virions. At this stage, we cannot exclude that the N-terminal extension additionally improves the enzymatic activity and/or Spike recognition of ZDHHC20$^{Long}$.

Spike S-acylation upon transfection into culture cells has been monitored by many laboratories[3,4,6,7,43,44], yet we all overlooked the fact that ZDHHC20$^{Short}$, which we now know was the main enzyme expressed in these cells, is rather inefficient in modifying the Spike population. This is likely because $^3$H-palmitic acid incorporation and acyl-capture assays monitor gains of signal such that abundant proteins with numerous cysteines, like over-expressed Spike, will produce readily detectable signals suggestive of extensive acylation. However, these methods do not inform on the stoichiometry of S-acylation. PEGylation experiments have been done, but the minimal band shifts of Spike that were observed were not interpreted[4,7,43,45], yet did indicate that acylation under these conditions was occurring on very few Spike molecules within the population. Instead, the Spike population as a whole becomes extensively acylated only when infection has

triggered the expression of significant levels of ZDHHC20$^{Long}$. During SARS-CoV-2 infection, cells require ≈24 h or more to acquire a significant population of ZDHHC20$^{Long}$ (Fig. 1d). During the first rounds of virus production, 6–12 h post infection, the produced virus will thus harbor significant non-acylated Spike proteins. What must follow is a period of extreme heterogeneity in Spike acylation, ranging from non-acylated trimers to trimers carrying 30 acyl chains. This heterogeneity might influence the size, lipid composition, and fusogenic capacity of the virions[4,5]. Spike is not the only viral surface glycoprotein modified by ZDHHC20. Another of its substrates is the hemagglutinin of the Influenza A virus[29]. It will be interesting to determine whether a similar change of enzyme and improved hemagglutinin acylation also operates during flu infection.

Our study opens intriguing questions both regarding Spike, and the damage response/repair pathway that leads to the expression of ZDHHC20$^{Long}$. For example, does acylation of a Spike trimer enhance its probability of being adequately incorporated into a nascent virus? Why do enveloped viruses carry so many glycoproteins, when only few are required for fusion? Is this to ensure that at least some are fully acylated, i.e., optimal for binding/fusion? Then more generally regarding ZDHHC20$^{Long}$ expression, why is it expressed in the small intestine? Is this part of the gut under permanent moderate mild stress that requires chronic repair? What do SARS-CoV-2 infection, loss of plasma membrane integrity through pore formation and chemically induced colitis have in common that leads to the transcriptional change of the *zdhhc20* gene? Our experiments on colitis and pore formation, suggest that ZDHHC20$^{Long}$ is preferentially expressed during the repair/recovery phase. How is ZDHHC20$^{Long}$ beneficial to these situations? While ZDHHC20$^{Long}$ enhances Spike fusogenic capacity, could ZDHHC20$^{Long}$ expression still be overall beneficial to the host during SARS-CoV-2 infection, counting in for example ZDHHC20-dependant antiviral responses[46]? Further studies addressing such questions hold the promise of interesting findings.

## Methods

### Ethics statement
For animal experimentation, all procedures were performed according to protocols approved by the Veterinary Authorities of the Canton Vaud and according to the Swiss Law (license VD 3497 and VD 3794, EPFL).

### Antibodies
ACE-2 (Abcam: ab15348; RRID: AB_301861; rabbit: 1:2000 dilution).

Actin (Millipore: MAB1501: RRID: AB_2223041; mouse: 1:4000 dilution).

Calnexin (Millipore: MAB3126: RRID: 2069152; mouse: 1:2000 dilution).

Climp-63/CKAP4 (Bethyl Laboratories: A302–257A; RRID: AB_1731083; rabbit: 1: 2000 dilution).

Flag (Sigma: F3165; RRID: AB_259529; mouse: 1:2000).

GAPDH (Thermofisher: 398600; RRID: AB_2533438; mouse: 1:4000 dilution).

Giantin (Abcam: ab37266; RRID: AB_880195; rabbit: 1:200 dilution).

GOLPH3 (Abcam: ab98023; RRID:AB_10860828; rabbit: 1:200 dilution).

GM130 (BD:610823; RRID: AB_3998141; mouse: 1:200 dilution).

HA (Roche: 11867423001; RRID: AB_390918; rat: 1:500 dilution).

Keratin-20 KRT20 (Cell Signalling: 13063; RRID: AB_2798106; rabbit: 1:600 dilution).

Myc (Sigma: M4439; RRID: AB_439694; mouse: 1:2000 dilution).

Nucleocapside N SARS-CoV-2 (Genetex: GTX135357; RRID: AB_2868464; rabbit 1:2000 dilution).

SARS-CoV-1/2 E and M antibodies are gifts from Machamer lab.

OLFM4 (Cell Signalling: 39141; RRID:AB_2650511; rabbit: 1:250 dilution).

Spike SARS-CoV-2 (Lifespan: LS-C19510; RRID: AB_840148; rabbit: 1:2000 dilution).

ZDHHC20 ALL (Sigma: SAB4501054; RRID: AB_10744838; rabbit: 1: 2000 dilution).

Mouse-HRP (GE Healthcare: NA931V; RRID: AB_772210; mouse: 1: 3000 dilution).

Rabbit-HRP (GE Healthcare: NA934V; RRID: AB_772206; rabbit: 1: 3000 dilution).

Mouse-Alexa488 (ThermoFisher Scientific: A-11029; RRID: AB_2534088; 1:800 dilution).

Mouse-Alexa568 (ThermoFisher Scientific: A-11037; RRID: AB_2534013; 1:800 dilution).

Rabbit-Alexa488 (ThermoFisher Scientific: A-21206; RRID: AB_2535792; 1:800 dilution).

Rabbit-Alexa568 (ThermoFisher Scientific: A-11042; RRID: AB_2534017; 1:800 dilution).

DAPI (ThermoFisher Scientific: D1306; RRID: AB_2629482; 1:5000 dilution).

Hoechst (Sigma: 94403; 1:5000 dilution).

## Polyclonal antibodies production against Long ZDHHC20[Long]
The two following peptides were used in combination to produce polyclonal antibodies in two different rabbits, two serum were pooled and were immunopurified against the two peptides (Peptide 1: C-EAGE LDQQPPGASES-coNH2- Peptide 2: C-LPRSPERWDAGLGSRQR-coNH2).

## Compounds and reagents
D-Biotin (Combi-Blocks: CSLSS-7910)

ENDO H (Bioconcept: P0702L)

Hydroxylamine (Sigma: 55460)

JQ1(Sigma: SML-1524; used final concentration 100 nM, FOXA1 inhibitor)

Mithramycin (LKT: LKT-M3476; used final concentration 100 nM, SP1 inhibitor)

NEM (Sigma: E1271)

Palmostatin B (Calbiochem: 178501; used final concentration 50 M, thioesterase inhibitor)

PEG-5KDa (Sigma: 63187)

TCEP (Sigma: C4706)

Zebra-spin desalting columns (Pierce: PIER89882)

3H-palmitic acid (American Radio-labeled Chemicals: ART0129-25)

35S-met-cys (Hartmann Analytic GmbH: IS103-185)

Dextran-Sulfate-Sodium Colitis grade (MP Biomediacals:160110).

## RNAscope probes for in situ hybridization
A mix of 7ZZ Probes tagged C3 were designed and produced by advanced cell diagnostic (ACD) in the following mouse ZDHHC20 sequence:

5′-GAGTCTTATATTTAAGTATATATAA- TATTTTTCTTGCTTGCACGCTTGAG TGATGCACAGCCTTATGTAATA GGGAACCCGACTTGTATGGGTCCTGAAA GACCTAGGGGAAGAGTGTT ATACAGAGTGCCACCCAAGGCACACAGGGCA GCTAAGAATTCATTC CTCTCCCCTTGCGCATCCCCATCCCTCTCGTCACC CAGTCCTAGAT GGCCTCCTACACATCCTTAGCACTCTCCTCTCTTTTCCA CCAAGGCA CCCCCAAATCCCATGGGCGGGCCTGAGCAGAAGCCCCGCCCC AACT TCAGGCCCCGCCTCCTTCGGCCGGGTAGCCCCCACCCCCTACGGGGA TTGCCAGGCGCGGGAC-3′.

RNAscope Multiplex Fluorescent V2 assay was performed according to the manufacturer's protocol 4-μm paraffin sections, hybridized with the probe described above at 40 °C for 2 h. The C3 channel channels were revealed with TSA Opal570 (Akoya Biosciences, Cat. No. FP1488001KT). After 30 min blocking with 1% BSA, tissues were incubated with the primary antibody: rabbit anti-Olfm4 overnight at 4 °C. The secondary antibody: donkey anti-rabbit Alexa488 was incubated for 45 min at RT before counterstaining with DAPI and coverslipping with Prolong Gold Antifade Mounting medium.

## Aerolysin purification and cell intoxication
Proaerolysin toxin was produced and purified by our laboratory in *Aeromonas salmonicida* as previously described[47]. Vero E6 were treated one hour in complete medium with 10 ng/ml of proaerolysin at 37 °C. Cells were washed twice in complete medium and further incubated at 37 °C for indicated times.

## Cell culture methods
Vero E6 (ATCC: CVCL_0574), HEPG2 (ATCC: HB_8065), Calu-3 (ATCC: HTB_55), HEK (ATCC: CRL_11268) cells were grown in DMEM Media supplemented with 10% fetal calf serum, 1% penicillin−streptomycin. HELA (ATCC: CVCL_0030) cells were grown in MEM Media supplemented with 10% fetal calf serum, 1% penicillin−streptomycin, 1% NEAA and 1% glutamine (all media from ThermoFisher Scientific). Parental HEK (ATCC: CRL_11268) cells and HEK293TphACE2-TMPRSS2[48,49] cells were kindly provided by Priscilla Turelli from Didier Trono Lab and cultured as standard.

## CRISPR/Cas9 deletion
Vero E6 KO CRIPR-Cas9 for ZDHHC20 was done in the background of Vero E6 WT with the following gRNA sequences to do a deletion in monkey *zDHHC20* gene:

gRNA(F): 5′-TGGCGTTAAGCTGATACCATTGG-3′

gRNA(R): 5′-CCATTGTGAAGTTAAGACATAGG-3′.

Vero E6 KO CRIPR-Cas9 for ZDHHC9 was done in the background of Vero E6 KOZDHHC20 with the following gRNA sequences to do a deletion in monkey *zDHHC9* gene:

gRNA(R): 5′-AGTCTATCCTATCAGCCCACCGG-3′,

gRNA(F): 5′-CATTGGGTCACATTTGCGGAAGG-3′.

HEK293T KO CRIPR-Cas9 for ZDHHC20 was done in the background of HEK293T WT with the following gRNA sequences to do a deletion in human *zDHHC20* gene:

gRNA (F): 5′-GCGTCCGAGTCACCGTCGCCGGG-3′

gRNA (R): 5′-ATTAAGGCATCATTCTGCTCTGG-3′.

## Protein extraction and western blot
For western blot, cells or epithelia or tissues were lysed 30 min at 4 °C in IP buffer (0.5% NP40; 500 mM tris-HCl, pH = 7.4; 20 mM EDTA; 2 mM benzamidine; 10 mM NaF and a cocktail of protease inhibitors), centrifuge 3 min at 2000×*g* and protein amount were quantified in supernatants and samples were processed for Western blot with the annotated antibodies.

## Plasmid constructs and overexpression

Plasmids were transfected in Vero E6 or HeLa cells for 24 h (3 μg/9.6 cm² plate using Transit-X2ᴿᵀᴹ (Mirus). For control transfection, we used an empty pcDNA6.2 plasmid.

Plasmids expressing WT or 10CA SARS-CoV-2 SPIKE were cloned in pcDNA6.2 with HA-tag in C-terminal for the initial plasmid (Addgene: 149329). Cystein to alanine substitution was done with Quickchange.

Plasmids expressing ZDHHC20$^{Long}$ and ZDHHC20$^{short}$ were cloned in pcDNA3.1 with myc-tag in N-terminal or with HA-tag in C-terminal. All deletions or mutations to Alanin were done by Quickchange.

Plasmids expressing reporter constructs ZDHHC20$^{Long}$ and ZDHHC20$^{short}$ and mutants were cloned in pcDNA3.1 with all amino acids sequence till the end of the first transmembrane domain: **M**WGGGGAWLGAGTGGVPAAPRTFAPTSPASEAGELDQQPPGA-SESPSPGLPR-SPERWDAGLGSRQRN**M**APWTLWRCCQRVVGWVPVLFFITFVVVW, followed by a linker: RILQSTVPRARDPPVAT and GFP sequence with N-glycolsylation site: MVSKGEELFTGVVPILVELDGDVNGHKFSVSGE-GEGDATYGKLTLKFICTTGKLPVPWPTLVTTFTYGVQCFAR-YPDHMKQHDFFKSAMPEGYVQERTIFFKDDGNYK-TRAEVKFEGDTLVNRIELKGIDFKENGSILGHKLEYNYNSHKVYI-TADKQKNGIKVNFKTRHNIEDGSVQLADHYQQNTPIGDGPVLLPDN-HYLSTQSALSKDPNEKRDHMVLLEFVTAAGITLGMDELYKstop. All deletion or mutations to Alanin in reporter constructs were done by Quickchange.

For RUSH experiments, ZDHHC20$^{Long}$ and ZDHHC20$^{short}$ were cloned in RUSH STR-Li-SBP-EGFP plasmids[25]. The following plasmids were purchased from Addgene: mScarlet-Giantin_C (85050), FOXA1-flag (153109); SP1-flag (25543). For stable expression of myc-ZDHHC20 constructs in cells HEK293T-KO-ZDHHC20, isolated cell clones were selected with 10 μg/ml of puromycin and assessed for equivalent level of expression.

## siRNA and silencing

Control siRNA, or specific human or Monkey siRNA listed below were purchased from Qiagen and transfected 72 h with 15 pmole/9.6-cm² plate using Transit-X2ᴿᵀᴹ (Mirus) as transfection reagent.

control: 5′-ATTGAACAAACGAAACAAGGA-3′
human/monkey ZDHHC9: 5′-CTCAACCAGACAACCAATGAA-3′
human FOXA1: 5′-CCAGACGGGTTTCATTATTAT-3′
human FOXA2: 5′-CACGTTCTATATAAGGAGGAA-3′
human SOX13: 5′-TTCACAAAGTTTGTTCCCTAA-3′
human RXRA: 5′-TTCGTGTAAGCAAGTACATAA-3′
human USF: 5′-CAGAGTAAAGGTGGGATTCTA-3′
human SP1: 5′-CAGCAAGTTCTGACAGGACTA-3′
human EID1: 5′-CTCGGCTGTGATGAGATTATT-3′
human GATA1: 5′-AAGCGCCTGATTGTCAGTAAA-3′.

## PEGylation-Acyl-PEG exchange

Acyl-PEG exchange was used to follow S-acylated protein by addition of mPEG to acylated cysteine following removal of hydroxylamine as previously[4]. Cell lysates were first incubated 30 min at RT with 10 mM TCEP. Free cysteine in cell lysates were blocked with 100 mM NEM, excess of NEM is removed by acetone precipitation. S-acylated cysteines were revealed by treatment 1 h at 37 °C with 200 mM neutral hydroxylamine. Lysates were desalted with Zeba spin columns and incubated 1 h at 37 °C with 2 mM 5 kDa methoxypolyethylene glycol maleimide. The addition is stopped by incubation with SDS-PAGE loading buffer with BME, and the lysate is analyzed by SDS-PAGE.

## Acyl-RAC

S-acylated proteins were purified using Thiopropyl Sepharose in the presence of 200 mM neutral hydroxylamine from cell lysates pretreated with 100 mM NEM to alkylate irrelevant cysteines[4].

## EndoH treatment

Following manufacturer instructions (NEB, P0702S), 40 μg of cell extract were denatured 10 min at 100 °C and treated 1 h at 37 °C with 1000 units of EndoH.

## 3H-palmitic acid incorporation-decay

HeLa cells were transfected with different constructs were incubated 1 h for starvation in IM (Glasgow minimal essential medium buffered with 10 mM HEPES, pH 7.4) and for indicated time in IM with 200 μCi/ml $^3$H-palmitic acid (9,10-$^3$H(N)) (American Radiolabeled Chemicals, Inc.). For decay analysis, after starvation, cells were incubated for 3 h in IM with 200 μCi/ml $^3$H-palmitic acid and cells were washed, incubated in DMEM complete medium for the indicated time of chase, or directly lysed for immunoprecipitation with the indicated antibodies.

For immunoprecipitation, cells were washed three times PBS, lysed 30 min at 4 °C in IP Buffer and centrifuged 3 min at 5000 rpm. Supernatants were subjected to preclearing with G sepharose beads prior immunoprecipitation reaction. Supernatants were incubated overnight with the appropriate antibodies and G Sepharose beads. After immunoprecipitation, washes beads were incubated for 5 min at 90 °C in reducing sample buffer prior to 4–20% gradient SDS-PAGE. Gels are incubated 30 min in a fixative solution (25% isopropanol, 65% H2O, 10% acetic acid), followed by a 30 min incubation with signal enhancer Amplify NAMP100 (GE Healthcare). The radiolabeled products were revealed using Typhoon phosphoimager and quantified using the Typhoon Imager (ImageQuanTool, GE Healthcare).

## 35S-methionine–cysteine pulse chase

HeLa cells were starved in DMEM HG devoid of Cys/Met for 30 min at 37 °C, pulsed with the same medium supplemented with 70 μCi/ml of $^{35}$S Cys/Met (American Radiolabeled Chemicals, Inc.) for 20 min, washed and incubated in DMEM complete medium for the indicated time of chase before immunoprecipitation as for 3H-palmitic acid radiolabeling experiments.

## Viral strains, Stock production, and titration with plaque-based assays

Passage 3 (P3) viral stocks were isolated from supernatants of Vero E6 cells, cultured in T75 flasks to a confluency of 80–90%, and infected at MOI ≈ 0.05 with passage-2 (P2) SARS-CoV-2 viral strain (lineage B.1) hCoV-19/Switzerland/GE-SNRCI-29943121/2020, (GISAID ID: EPI_ISL_414019). P2 working Delta or Omicron stocks, were also produced in Vero E6 from Delta B1.617.2 (EPI_ISL_1811202) or Omicron BA.1 (EPI_ISL_7605546) isolates. Infected supernatants containing viral stocks were harvested between 48 and 72 h post inoculation, -10 ml of DMEM supplemented with 2.5% FCS or Eppiserf serum-free media. Supernatants were clear of cell debris by centrifugation (500×g 10 min) and filtration (0.45 μm), aliquoted, and stored at −80 °C. Viral titer was quantified by determining the number of individual plaque-forming units after 48 h of infection in confluent Vero E6 cells. In brief, viral stocks were serially diluted (tenfold) in serum-free medium and (400 μl) inoculated in triplicate 48 wells, confluent Vero E6 cells (2.5 × 10⁵) cells per well. After 1 h, inoculums were discarded, and cells were overlaid with a mixture of 0.4% of Avicel-3515 (Dupont) (from 2% stock) in DMEM supplemented with 5% FCS and penicillin and streptomycin for an additional 48 h. Overlays were discarded, and cells were fixed in 4% paraformaldehyde (PFA) for 30 min at RT. Fixed cells were washed in PBS and stained with 0.1% crystal violet solution (in 20% ethanol/water) for 15 min. The staining solution was discarded, and the wells were washed twice in water. Plates were allowed to dry and analyzed for quantification of the cytopathic effect as the number of individual Plaque-forming units (PFU) per ml (Avg PFU*1/volume*1/dilution factor). Such un-concentrated viral stocks yielded between 0.5 to 5 × 10⁶ PFU/ml.

## SARS-CoV-2 infections

Unless otherwise indicated, all infections were done using P3 SARS-CoV-2 (B.1) stocks. P2 stocks were used for experiments using the Omicron and Delta variants. Vero E6, Calu-3 cells or HEK293T seeded to a confluency of 90 to 100% were, washed twice in warm serum-free medium and inoculated with the indicated MOI of SARS-CoV-2, diluted in serum-free medium (5 ml for a T75 and 2 ml for T25 flask—stocks and biochemistry experiments, and 500 µl for 12-well plates—infectivity assays). 1 h after inoculation, cells were washed with complete medium, and infection was allowed to proceed for the indicated time points in DMEM supplemented with 2.5% FCS, penicillin, and streptomycin (10 ml for T75; 4 ml for T25, and 1 ml for 12-well plates). Infected cells and virion supernatants were harvested and inactivated in IP buffer (1:1 v/v for supernatants) or lysed in Pegylation/Acyl-Rac lysis buffer (0.5% Triton-X100, 2.5% SDS, 25 mM HEPES, 25 mM NaCl, 1 mM EDTA, pH 7.4, and protease inhibitor cocktail) for at least 30 min.

## Titration of viral E and N copies

For titration of viral RNA, equivalent volumes of RNA extracted from infected culture supernatants or RNA from serial dilutions of SARS-CoV-2 (N + E) RNA Quant Standard (Cat.# AM2050 Promega) were used for cDNA synthesis. Samples were used to perform QPCR analysis (using E and N-specific primers). A standard curve was generated by plotting the Ct values against the number of E/N copies per µl indicated by the serial dilution of the E/N standards (stock at $4 \times 10^6$ copies/µl) and used to extrapolate the number of Viral N/E copies/ml in samples.

## SARS-CoV-2 single-round infectivity analysis

KO-ZDHHC20/9 cells cultured in T75 flasks transfected with empty plasmids or ZDHHC20 (Short or Long) expressing plasmids (24 h) were infected as described above. Supernatants were harvested between 24 and 48 h post inoculation and processed for QPCR analysis (150 µl) or aliquoted and stored at −80 °C until titration of viral N/E copies. After titration and adjustment to viral N/E RNA copies nonconcentrated SARS-CoV-2 supernatants were used to infect confluent Vero E6 cell monolayers cultured in 12-well plates using an approximate ratio of 5 to 10 N/E copies per cultured host cell. At least three wells were infected per supernatant per condition. Infection was done as described until 6 h post inoculation when cells were washed lysed and processed for QPCR analysis. Infected cells were harvested and lysed in 330 µl of Maxwell® RSC Viral Total Nucleic Acid Purification Kit-lysis buffer from Promega, incubated at 80 °C for 10 min, and used for Viral RNA extraction according to manufacturer's instructions. RNA concentration was measured, and 500 ng or 1000 ng of total RNA was used for cDNA synthesis using iScript. A 1:5 dilution of cDNA was used to perform quantitative real-time PCR (QPCR) as described below. Ct values from primers used for viral genes (E, RdRp, and N) were used for quantification of N/E copies using SARS-CoV-2 RNA standards as described or normalized to host housekeeping genes (ALAS-1, RPL27) for cell replication infectivity assays. Results were expressed as 2^(-ΔΔCt)*100%.

## VLPs production

HiBiT-N-tagged virus-like particles (VLPs) were produced as described previously[4,50]. Briefly, equimolar amounts of full-length CoV S, E (envelope), M (membrane), and HiBiT-N-encoding plasmids (total, 10 µg) were transfected into KO-ZDHHC20 HEK293T cells stably expressing plasmids coding for ZDHHC20 (Short or Long) or empty plasmids) and co-transfected with siRNA targeting ZDHHC9 for 72 h. To produce Bald ("no-S") VLPs, the S expression plasmids were replaced with empty vector plasmids. At 6 h post-transfection, cells were replenished with fresh DMEM-10% FBS. HiBiT-N VLP suspensions were collected in FBS-free DMEM or Eppiserf from 24 to 48 h post-transfection. Suspensions containing HiBiT-N VLPs were clarified by

centrifugation (300×g, 4 °C, 10 min; 3000×g, 4 °C, 10 min). To obtain purified viral particles, clarified VLP suspensions were concentrated 100-fold by overlaid onto 20%, wt/wt, sucrose cushions and particles purified via slow-speed pelleting (SW32 8000 rpm, 4 °C, 20 h). The resulting pellet was resuspended in Eppiserf to 1/100 of the original medium volumes. VLPs were stored at −80 °C or analyzed promptly for titration using Nano-Glo® HiBiT Extracellular Detection System (Promega #N2420) with passive lysis buffer (Promega #E1941).

## VLP cell entry assay

HEK293TphACE2-TMPRSS2 target cells, cultured in 96-well plates precoated with poly-lysine, were transfected with pcDNA3.1-hACE2-LgBit. At 2 days post-transfection, cells were incubated with a live-cell Nluc substrate (Nano-Glo Vivazine; Promega), and 2 h later, cells were incubated at 4 °C on ice. HiBiT-N VLPs were inoculated at equivalent N-HiBiT input multiplicities, using four independent wells per VLP stock replicate. Nluc levels were quantified immediately and set as $T = 0$, before incubation of the plates at 37 °C. HiBiT-N VLPs lacking Spike proteins (bald) served as negative controls. At the indicated intervals following VLP inoculation, Nluc levels were quantified using a HIDEX microplate reader. For data presentation, the Nluc recordings in cultures inoculated with bald VLPs were normalized to values of 1.0, and the fold increases over this control condition were calculated and plotted as "relative entry."

## Human MucilAir viral assays

MucilAir-human upper respiratory tissues (Epithelix, Geneve, Switzerland) were maintained according to the manufacturer's protocol. Prior to infection, tissues were washed apically with 200 µl of DPBS, calcium, magnesium (#14040091—Gibco) for 20 min at 37 °C and the basal medium replaced with fresh mucilair medium. Tissues were infected at the MOI of 0.1 PFU (assuming the manufacturer's estimations of 500000 cells per tissue). Tissues were inoculated with 200 µl of SARS-CoV-2 (passage 3) for 3 h (apically) at 33 °C. The apical inoculum was removed and infected tissues were maintained for until harvest. Harvest was performed using 300 µl lysis buffer for 30 min. Lysed inactivated tissue lysates were transferred to clean tubes and processed for western blot.

## hK18-ace2 mice virus infection

In total, 20 female and 6 male 11-week-old male and female K18-hAce2 C57BL/6j transgenic mice (strain: 2B6.Cg-Tg(K18-ACE-2)2Prlmn/J) from The Jackson Laboratory were used. Four females were used as uninfected controls and the remaining animals (16 females and 6 males) were administrated intranasally $1 \times 10^3$–$10^4$ PFU SARS-CoV-2 (B.1). Mice were euthanized between 1 and 7 days of infection and immediately dissected for organs collection and further analysis. Proteins were extracted as described above and viral and tissue RNA were extracted following instructions of Omega Bio-TEK, (EZNA total RNA KIT I). Procedures were performed according to protocols approved by the Veterinary Authorities of the Canton Vaud and according to the Swiss Law (license VD3794A, EPFL).

## DSS-induced mice colitis

Six wild-type C57BL/6j mice (8-week-old males) were used. Three ($n = 3$) mice were used as controls and three mice were given 3% dextran sulfate sodium in the drinking water for 7 days, then switched to regular drinking water for 3 days. Three Control 8-week male were given drinking water that did not contain DSS. During the 10-day experiment, mice were weighted and disease activity index (DAI) was performed daily based on stool consistency, presence of occult blood and body weight loss. On day 10, mice were euthanized with injection of pentobarbital and bled via cardiac puncture, and colon and intestines were collected. For animal experimentation, all procedures were performed according to protocols approved by the Veterinary

Authorities of the Canton Vaud and according to the Swiss Law (license VD 3497, EPFL).

## Quantitative real-time PCR

RNA was extracted from cells using Maxwell RSC Promega Kit (AS1330) and from mice tissues using E.Z.N.A total RNA kit (R6834). RNA concentration was measured and 500 ng of total RNA was used for cDNA synthesis using iScript (Biorad: 1708891). A 1:5 dilution of cDNA was used to perform quantitative real-time PCR using Applied Biosystems SYBR Green Master Mix (Thermofisher Scientific) on 7900HT Fast QPCR Applied Biosystems with SDS 2.4 Software.

The data in triplicate were normalized to housekeeping genes (human: ALAS-1, GUSS, TBP; mouse: RSP9, EEF1A1, COX6a1). Results were expressed as 2^(-DDCt)*100%.

Primers (all 5′ to 3′) used were:

Housekeeping gene:

Human GUSS: F: CCACCAGGGACCATCCAAT; R: AGTCAAAAT ATGTGTTCTGGACAAAGTAA

Human TBP: F: GCCCGAAACGCCGAATATA; R: CGTGGCTCT CTTATCCTCATGA

Human ALAS-1: F: CTCACCACACACCCCAGATG; R: AGTTCCAGCC CCACTTGCT

Human RPL27: TGTCCTGGCTGGACGCTACT; CTGAGGTGCCA TCATCAATGTT

Mouse RSP9: F: GACCAGGAGCTAAAGTTGATTGGA; R: TCTTGG CCAGGGTAAACTTGA

Mouse EEF1A1: F: TCCACTTGGTCGCTTTGCT; R: CTTCTTGTC CACAGCTTTGATGA

Mouse COX6a1: F: CTCTTCCACAACCCTCATGTGA; R: GAGGCCA GGTTCTCTTTACTCATC

Human ZDHHC20:

Human C1: F: CGCCTTGTGGGGATGGATCC; R: CCGCTTGTTGG ACAGTGAATCTCAG

Human C2: F: GTCGTCTGGTCCTACTACGC; R: AGCCACAA GGTAAACAACGG

Human 1: F: GTGGCCTTATTTGGAAGTGGG; R: CCCTACTCTA GTATGGCCTC

Human 2: F: GCCACTGCCACTGCAGGTC; R: CACAGCCATGTG CCCTCTG

Human 3: F: GGCACAAAATCAGGGAGAACAG; R: AAGAAG ACGTCACCCTTTGCT

Human 4: F: AGTCTCCCTCCCCTATTGAGT; R: AAAACACGC CCTTGATGGAT

Human 5: F: CCTCGGACTTTTGCTCCCACAAG; R: CGTCCCACCGT TCTGGGGAG

Mouse ZDHHC20:

Mouse C1: F: ATATTGCCTTTTTGTGGCTGC; R: ACTGTTGGTT CATTCGTCCAA

Mouse C2: F: TCATCACTGTCCATGGGTGAA; R: ACTGTTGGTT CATTCGTCCAA

Mouse 5′UTR1: F: CCTCTTCCTCCTGAGTGTGTGG; R: CCCAGGG GTTGTCTGAGGACA

Mouse 5′UTR2: F: CGCCCTTCGCCACTGCTTG; R: CTGGGGTCTTG TGGTCCTAC

Human and monkey transcription factors:

FOXA1: F: TGGAACAGCTACTACGCAGAC; R:GGTGTTCATGGTCA TGTAGGT

SP1: F:GCCACCATGAGCGACCA; R: GAAAAGGCACCACCACCATT

EID1: F: TCGTCTGACCGAAGAACTCG; R: TGGGTCCCTCCTCA AGTAGT

SOX13: F: CTCCAGAGGGTAATGGGTCC; R: CTATGGCTGGCAC CACTTCT

USF1: F: CCTTGGATAGGAAAGGACTTAGC; R: ATCTGCACTG TCCCCTCTTC

RXRA: F: CAGCGGAACCAAAACTGCT; R: GGTGAGCTGAGCCGGT

Mouse transcription factors:

FOXA1: F: ACTCTCCTTATGGCGCTACC; R: ACACCTTGGTAGT AGGCTGG

SP1: F: GTGGGAAGCGCTTTACACGTTCGG; R: GCCTGCCCTGAG TGCCCTAAG

SOX13: F: CCCGACCGATTAGATGTCCA; R: GCAAGGCTCCTTCT TCTCCT

USF1: F: ATCCAAAGACGGAGAAGGCT; R: GAATGCTAAGTCC GGGCCA

RXRA: F: GCAGACATGGACACCAAACA; R: CACCTGGGTAGAG AAGTCGAG

SARS-CoV-2:

E SARBECO: F: ACAGGTACGTTAATAGTTAATAGCGT; R: ATAT TGCAGCAGTACGCACACA

RDRP: F: AGCTTGTCACACCGTTTC; R: AAGCAGTTGTGGCATCTC

N (Nucleocapsid): F: GACCCCAAAATCAGCGAAAT;R: TGTAGC ACG ATTGCAGCATTG.

## 5′-rapid amplification of the cDNA ends (5′-RACE)

We applied SMARTer 5′-RACE technique using independent RNA preparations extracted from Calu-3 cells infected with SARS-CoV-2 for 24 h at MOI 0.1. A modification of the manufacturer's protocol (Clontech) was carried in order to generate first-Strand cDNA Synthesis from specifically *zdhhc20* transcripts. For that, instead of using the modified oligo (dT) primer, cDNA synthesis was done using a *zdhhc20*-specific reverse primer: R:GTA CGC GTA GTA GGA CCA GAC. 5′-RACE was then carried with the gene (*zdhhc20*) specific primer #4 modified for infusion cloning. 5′-GAT TAC GCC AAG CTT CGT CCC ACC GTT CTG GGG AG-3′. 5′-RACE products were loaded in an agarose gel and a single large fragment around >9000 bp was excised for DNA purification and In-Fusion cloning (Clontech). A total of 53 transformed clones were confirmed as *zdhhc20* 5′ ends by Sanger sequencing using *zdhhc20* primer #4. A summary of the sequenced mRNA species and their frequency is shown in Supplementary Fig. 1D and Supplementary Table S2.

## Immunohistochemistry

Vero E6 or HeLa cells seeded in glass coverslips in 24-well plates and transfected with the indicated myc-/3HA-/GFP-ZDHHC20 constructs (for 24 h) were fixed in 4% paraformaldehyde (15 min), quenched with 50 mM NH4Cl (30 min) permeabilized or not with 0.05% Saponin (5 min). Antibodies were diluted in PBS containing 1% BSA and 0.05% Saponin for permeabilization. Coverslips blocked in PBS 1% BSA, for 30 min, washed and incubated with primary antibodies overnight at 4 °C, washed three times in PBS, and incubated 45 min to 1 h with secondary antibodies and when indicated nuclear staining Hoechst. Coverslips were mounted onto microscope slides with ProLong™ Gold Antifade Mountant. For exclusive imaging of GFP-expressing reporters, coverslips were nuclear-stained and mounted without permeabilization. Images were collected using a confocal laser-scanning microscope (Zeiss LSM 700) and processed using Fiji™ software.

## Rush

Cells were seeded in 35-mm glass bottom plates (FD35-PDL, FluoroDish™) and transfected with 1 μg of each plasmid DNA encoding for ZDHHC20-RUSH reporter and Scarlet-Giantin. RUSH reporters encode for both a GFP-ZDHHC20 fusion tagged with streptavidin-binding protein (luminal) and the ER-resident hook—the transmembrane invariant chain (Ii)—streptavidin-tagged. After 48 h of transfection, cells were washed and culture medium replaced with pre-warmed carbonate-independent Leibovitz's medium (Invitrogen). Time-lapse imaging was performed on a Visitron Spinning Disk CSU W1 with full temperature control and $CO_2$, under culture conditions. Z-stack (0.7-μm slice) images were acquired for each channel before and after the addition of Bio-D ($T = 0$) at final concentration 40 μM, every 10 s for at

least 1800 s (30 min). Acquisitions were processed using Fiji™ software 45. The mean fluorescence intensities were measured for each time-lapse image, for two ROIs: Total cell (marked by initial GFP-ZDHHC20 staining) and Golgi (marked by Scarlet-Giantin). Golgi/total fluorescent ratios were normalized to 1 for $T = 0$ and corrected values plotted using GraphPad Prism.

## Automated microscopy analysis

Approximately 10,000 HeLa MZ cells were plated in FluorobriteTM in 96-well imaging plates from Ibidi (ref: 89626) for 24 h. Cells were transfected for 24 h with 1 μg/mL final concentration of different PAT20 plasmids using Mirus Transit-X2 transfection reagent at a final dilution of 1/1000. Cells were fixed with 3% PFA, permeabilized with 0.05% saponin and blocked with 1% BSA. Cells were then stained with anti-GM130 and anti-MYC (when necessary) antibodies both at 1/200 followed by incubation with secondary antibodies tagged with Alexa-fluor 647 and 488 (for MYC staining) with Hoechst at 1/2500. In all conditions, at least nine images per well were acquired using IXM confocal automatic microscope (Molecular Device) using ×20 water immersion objective 0.95NA. Analysis of the automated microscopy images was done via the MetaXpress Custom Module editor software. The images were segmented to generate relevant masks, which were then applied on the fluorescence images to extract the relevant measurements. Cell and Nuclei masks were created using Hoechst to generate the master object (cell). To facilitate segmentation of Golgi, we applied the top hat deconvolution method, to reduce the background noise and highlight bright objects. A logical operation was used to generate the mask of cytoplasm without Golgi. Relevant masks were then applied to the fluorescent images to extract relevant measurements.

## Phyml tree

A list of orthologs from all species for the human gene ZDHHC20 using the ensembl.org website was established. These transcripts were used as query sequences to a blastn/megablast[51] search to retrieve RefSeq and EMBL/Genbank transcripts with very high similarity.

Those retrieved transcripts were analyzed using a Perl script to find potential 5′ extension of the known ORF until the 5′ most potential TSS while staying in-frame for the transcription. All the retrieved transcripts (both those where an extension is possible and those where it is not) were translated to the corresponding amino acid letter codes and aligned using mafft into a multiple sequence alignment (MSA)[52]. The MSA was hand-edited to remove duplicates and select one representative sequence per broad taxonomic group. The taxonomic information was retrieved from the corresponding RefSeq and/or EMBL/Genbank entries. The resulting MSA was provided as input to phyml[53], which was requested to perform 100 cycles of bootstrap. The resulting output tree was plotted using the ape R package[54] to produce a PDF document that was used to generate the Figure.

## UCSC genome browser analysis

UCSC genome browser at ZDHHC20 locus was obtained from (Human version hg19). Two custom tracks and three UCSC genome browser tracks were selected for display in Fig. 1 and Supplementary Fig. 1. Transcript annotation corresponds to GENCODE Genes track (version V4 0lift37); Chromatin State displays chromatin state segmentation data from ENCODE consortia for two different cell lines (H1.hESC and HepG2) Transcription Factor CHIP-seq Peaks track shows transcription factor (TF) binding sites for HepG2 cell line based on ChIP-seq experiments from ENCODE. Only TF with a minimum score range of 300 are shown. The level of peak enrichment is represented with the darkness of the item, and the vertical pink bar marks the point-source of the peak. TFs that have been experimentally tested in this study are shown in pink. Data from murine *zdhhc20* locus was obtained from

Mouse GRCm39/mm39 from UCSC genome browser https://genome.ucsc.edu; Data from murine transcripts was obtained from NCBI RNA reference sequences-Refseq and GENCODE (version VM30) and mRNA sequences from Genbank.

## Reproducibility and statistical analysis

Unless otherwise indicated, all data (e.g., from western blots, autoradiography images, immunofluorescence images, and quantifications) were repeated at least three times independently. Unless otherwise stated, each data point corresponds to one biologically independent experiment/replicate with consistent results. Statistical analysis was carried out using Prism software. Data representations and statistical details can be found in the description of the Figures. For ANOVA analysis, $P$ values were obtained by post hoc tests to compare every mean and pair of means (Tukey's & Sidak's) or to compare every mean to a control sample (Dunnet's). Estimated half-lives, from metabolically labeling experiments were extracted from the individual experiments using non-linear regression with one-phase decay (Prism).

## Reporting summary

Further information on research design is available in the Nature Portfolio Reporting Summary linked to this article.

## Data availability

All data supporting the findings of this study are available within the paper, the Supplementary Information and Supplementary Data Source files. All data regarding the human *zdhhc20* locus was obtained from the Genome browser on Human (GRCh37/hg19) (version hg19); Transcript data obtained from GENCODE Genes track (V4 0lift37) https://genome.ucsc.edu/cgi-bin/hgTrackUi?g=knownGene; Chromatin State segmentation obtained from ENCODE https://genome.ucsc.edu/cgi-bin/hgTrackUi?db=hg18&g=wgEncodeBroadHmm. All data from murine *zdhhc20* locus was obtained from Mouse GRCm39/mm39 from UCSC genome browser https://genome.ucsc.edu. Source data are provided with this paper.

## Code availability

All details on methods (e.g., references, databases and R packages) used for the generation of the ZDHHC20 Phylm tree (Fig. 1l) can be found in the method section. All details regarding the code for used for RNAscope quantification in Figs. 2, 3, and Supplementary Fig. 3 are provided as Supplementary information and are available in GitHub (https://github.com/upvdg/rnascope-qupath).

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

## Acknowledgements

We thank C. Iseli and N. Guex for ZDHHC20 Phyml tree realization.; BIOP EPFL Core Facility; S. Vossio and D. Moreau from ACCESS Geneva; P. Turelli, EPFL for the different viral strain production; F. Perez and G. Boncopain for RUSH plasmid constructs; Machamer laboratory for E and M. antibodies[55]; all G. Van der Goot lab and D. Trono lab members for discussions. This work was supported by the Swiss National Science Foundation (SNSF-31CA30_196651) and by CARIGEST S.A.

## Author contributions

Conceptualization: F.S.M., L.A., and F.G.v.d.G.; investigation: F.S.M., L.A., B.K., L.B., N.P., V.M., A.C., and J.C.-F.; funding acquisition: F.S.M. and F.G.v.d.G.; writing—original draft: F.S.M., L.A., and F.G.v.d.G.; writing—review and editing: F.S.M., L.A., L.B., V.M., J.C.-F., D.T., and F.G.v.d.G.; resources: B.K., A.C., and L.A.

## Competing interests

The authors declare no competing interests.
