## [Peer Review File · Nature Communications]

Reviewers' Comments:

Reviewer #2:

Remarks to the Author:

The revisions refocus the paper on transcriptional regulation of ZDHHC20 by SARS-CoV-2 infection and other stress/cell damage responses and its impact on Spike acylation. The significance of the report is not diminished by this shift in emphasis. The work opens and inspires new lines of investigation both in SARS-CoV-2 infection biology and the S-acylation field.

The authors have satisfactorily addressed the concerns in my review and those raised by the other reviewers. Where appropriate, given the change in focus of the manuscript, the authors responded with additional data and revisions of the text, resulting in an improved manuscript.

My one remaining concern is the new title, as "transcriptionally improved" lacks a clear meaning. A suggestion -

SARS-CoV-2 hijacks a cell damage response to induce transcription of a more active Spike S-acyltransferase.

Reviewer #3:

Remarks to the Author:

In this study Mesquita et al. report that SARS-CoV-2 infection induces expression of a longer isoform of the ZDHHC20 acyltransferase. This results in more efficient acetylation of the Spike protein, a modification that is important for fusogenic activity. They show this expression is due to FOXO1 and SP1 and that it also occurs in other settings where there is cellular damage. The longer protein isoform is more stable and more active on Spike, potentially because it is localized to the ER and not the Golgi. They propose SARS-CoV-2 is inducing a cellular response to take advantage of the acylation. I previously reviewed this manuscript for a different nature journal, and as before, I find it to be a thorough and comprehensive study. The reported finding is interesting and it has implications not only for SARS-CoV-2 but also for other stress responses. In this revision, they have addressed all my main points, reducing the emphasis on the connection to SARS-CoV-2 biology and explaining their other stress challenges more in detail. I have a few more minor writing comments.

1) Lines 132-148 and 585-604 are more discussion than results. Reading flow would be greatly improved by moving those paragraphs and integrating them with the conclusions section.

2) When talking about the effects of SARS-CoV-2 infection on ZDHHC20 isoform expression in different mouse tissues (lines 245-261), the authors should explicitly say that some of the changes may be due to paracrine signaling, since SARS-CoV-2 proteins are not detected in some of the tissues that show ZDHHC20 isoform shifts. (or if the authors have another explanation for this observation, that should be made explicit). In particular on line 261 and line 321 "all SARS-CoV-2 infected tissues/organs" should be revised to say "all tissues/organs in SARS-CoV-2 infected mice".

3) Their results with ZDHHC20 expression in the DSS-treated colon (and to some extent aerolysin-treated Vero cells) suggest that long ZDHHC20 comes up more strongly during the recovery rather than the damage phase of the stress treatment. It would be good if the authors commented on this idea and what it may mean for the role of ZDHHC20.

4) The mutational analysis in lines 405-417 is hard to follow. Instead of using the T1-4 monikers,

could the authors simply state the deletion e.g. delta aa1-x.

5) Lines 423-424 – this phrase is convoluted, and I was confused when reading it. Should be changes to “Endoglycosidase H (Endo-H), which cleaves simple N-linked sugars generated in the ER, but not the complex ones generated in the Golgi”.

6) Lines 434-439 – could the authors add a one-sentence explanation of what they would see if ZDHHC20 was in the ER via retrieval?

7) Lines 570-573 – this statement re: modified cysteine per molecule seems contradictory to previous ones in the manuscript, which imply that they are all modified once one is.

8) Maybe I missed it but I could not find an explanation/reference to Extended data figure 5c.

Reviewer #4:

Remarks to the Author:

Re: SARS-CoV-2 hijacks a cell damage response pathway resulting in a transcriptionally 2 improved S-acyltransferase for Spike.

The authors have published S-acylation of Spike protein in Dev Cell (2021). This manuscript is describing post-translational modifications of Spike protein after SARS-CoV-2 infection. SARS-CoV-2 infection augments the S-acylation of S protein by inducing host S-acyltransferase, ZDHHC20Long, which is N-extended ZDHHC20 protein. This protein shows higher S-acylating activity to Spike protein in ER, resulting in high fusogenic SARS-Cov-2 viruses for better cell entry. The manuscript is well described.

1. Lines 58-9. As the damage response pathway, the gain of damage response is mainly for the host. When the expression of ZDHHC20Long increased, it played a beneficial role to the virus rather than to the host. It is necessary to consider how it plays a role in helping the host.

2. In Fig 4k, the smear pattern of ZDHHC20 could be observed. Why, however, are the smear patterns of ZDHHC20 differently shown?

3. ER retention motif of 'PERW' is different from previously published motifs. Please describe the molecular interpretation of the mechanism by which this is retained in the ER.

4. It seems that the explanation in the figure legend of Figure 4 j and l has changed.

5. Graphic summary in last figure is strongly recommended to include.

POINT-BY-POINT REPLY TO THE REVIEWERS

Reply to Reviewer #2

We are very happy to read that the reviewer feels that the “significance of the report is not diminished by this shift in emphasis” and that the manuscript has been improved by the additional data and revisions of the text.

“My one remaining concern is the new title, as “transcriptionally improved” lacks a clear meaning. A suggestion - SARS-CoV-2 hijacks a cell damage response to induce transcription of a more active Spike S-acyltransferase”.

The title has been changed: *SARS-CoV-2 hijacks a cell damage response, which induces transcription of a more efficient Spike S-acyltransferase*

Reply to Reviewer #3

We are very happy that the referee considers that our responses fully addressed their main concerns. We thank the reviewer the suggestions to improve the manuscript.

We have highlighted all the changes we have made in the manuscript in yellow. The text that is highlighted in green corresponds to existing text that we have move to the discussion section following your suggestion below.

1) Lines 132-148 and 585-604 are more discussion than results. Reading flow would be greatly improved by moving those paragraphs and integrating them with the conclusions section.

Thank you for this excellent suggestion, which has significantly improved the flow of the manuscript. The “Conclusion” has been converted to “Discussion”.

2) When talking about the effects of SARS-CoV-2 infection on ZDHHC20 isoform expression in different mouse tissues (lines 245-261), the authors should explicitly say that some of the changes may be due to paracrine signaling, since SARS-CoV-2 proteins are not detected in some of the tissues that show ZDHHC20 isoform shifts. (or if the authors have another explanation for this observation, that should be made explicit). In particular on line 261 and line 321 “all SARS-CoV-2 infected tissues/organs” should be revised to say “all tissues/organs in SARS-CoV-2 infected mice”.

We agreed with the reviewer. We now mention potential paracrine signalling and have made the suggested modification.

3) Their results with ZDHHC20 expression in the DSS-treated colon (and to some extent aerolysin-treated Vero cells) suggest that long ZDHHC20 comes up more strongly during the recovery rather than the damage phase of the stress treatment. It would be good if the authors commented on this idea and what it may mean for the role of ZDHHC20.

We thank the referee for bringing this point up. We agree with the comment. The DSS data indeed suggests that ZDHHC20^{Long} comes up during the regeneration stage, as we now mention. Also, ZDHHC20^{Long} comes up quite some time after the transient exposure of cells to aerolysin, but not, as shown in extended data Figure 3f, upon continuous exposure to the

toxin, again suggesting that the transcriptional shift might be part of the recovery stages in cellular responses to damage, as now mentioned.

Future studies are required to determine why ZDHHC20^{Long} expression is useful to the cell or the gut. With this perspective, it might seem contradictory that ZDHHC20^{Long} is useful to Spike during SARS-CoV2 infection. We have now clarified this point. Our findings indicate that this response pathway has been hijacked to optimise Spike acylation and thereby viral fusion. This does not exclude that ZDHHC20^{Long} expression could have consequences through other targets, such as antiviral proteins as mentioned at the end of the discussion.

4) The mutational analysis in lines 405-417 is hard to follow. Instead of using the T1-4 monikers, could the authors simply state the deletion e.g. delta aa1-x.

We have now simplified the names of the deletion mutants as suggested.

5) Lines 423-424 – this phrase is convoluted, and I was confused when reading it. Should be changes to “Endoglycosidase H (Endo-H), which cleaves simple N-linked sugars generated in the ER, but not the complex ones generated in the Golgi”.

We have changed the sentence as suggested.

6) Lines 434-439 – could the authors add a one-sentence explanation of what they would see if ZDHHC20 was in the ER via retrieval?

This point indeed required more explanation. First, we have changed the order of our panels, presenting the RUSH experiments before the EndoH sensitivity. We now make clear that in our confocal and live imaging experiments, we never see a Golgi pool of ZDHHC20^{Long}, which we would have expected to see considering the time frame of the RUSH experiments, and the speed at which images were taken. The movies indicate that ZDHHC20^{Long} does not leave the ER, i.e. is retained. To obtain independent evidence addressing retention vs. retrieval, we analysed EndoH sensitivity. The absence of complex sugars indicates that the protein has not reached the Golgi. So jointly, these experiments indicate that ZDHHC20 isoforms containing the PERW motif are retained in the ER.

7) Lines 570-573 – this statement re: modified cysteine per molecule seems contradictory to previous ones in the manuscript, which imply that they are all modified once one is.

Throughout the description of Fig. 5e, we have now made clear that the number of modified cysteines is expressed at the population level: so 1 cysteine modified in all Spike molecules would give the same average as 10% of the Spike population having all its 10 cysteines modified, which is what the data from our previous and current study support.

8) Maybe I missed it but I could not find an explanation/reference to Extended data figure 5c.

We have now added a reference to Extended data figure 5c, which illustrates that the relative abundance of structural SARS-CoV2 proteins, Spike, N and M was not significantly influenced by which acyltransferase was expressed

Reply to Reviewer #4

We are pleased that the reviewer has appreciated the manuscript and how it was written.

We have highlighted all the changes we have made in the manuscript in yellow. The text that is highlighted in green corresponds to existing text that we have move to the discussion

section following the suggestion of Reviewer 3, which indeed significantly improves the flow of the manuscript.

1. Lines 58-9. As the damage response pathway, the gain of damage response is mainly for the host. When the expression of ZDHHC20^{Long} increased, it played a beneficial role to the virus rather than to the host. It is necessary to consider how it plays a role in helping the host.

The reviewer rightly points to an issue that we had not explicitly mentioned in the text and now do. This point was also brought up by reviewer 3. The fact that ZDHHC20^{Long} is expressed upon transient exposure to pore-forming toxins and during the recovery phase of DSS-induced colitis led us to hypothesize that its expression is part of a damage response pathway, which should in principle indeed be good for the host and future study will need to determine how this pathway helps the host. In the case of SARS-CoV2, the pathway is hijacked to benefit Spike. But that does exclude that the damage response pathway could overall be beneficial to the virus. Our study is focused on Spike and its role in infection.

This is now mentioned for each experimental set up (colitis, pore formation) and at the end of the discussion.

2. In Fig 4k, the smear pattern of ZDHHC20 could be observed. Why, however, are the smear patterns of ZDHHC20 differently shown?

We are not sure we have understood what the reviewer means by “differently shown”. The migration patterns of these reporters are complexes for reasons that we do not understand and have not addressed. This is now mentioned. The multiple bands might be due to partial degradation, partial hydrolysis or other reasons. We have focused on the sensitivity of the highest molecular weight, and most abundant, forms to Endoglycosidase H (Endo-H). ZDHHC20^{Short} and some of the deletion mutants show an upper smeary pattern, characteristic of the presence of complex N-linked sugars, while ZDHHC20^{Long} and the rest of the deletion mutants show well-defined highest molecular weight bands. Consistent with the appearance (smear vs. sharp band), the smears remained upon EndoH treatment, while the sharp bands migrated at a lower molecular weight.

3. ER retention motif of ‘PERW’ is different from previously published motifs. Please describe the molecular interpretation of the mechanism by which this is retained in the ER.

We can only speculate. The simplest interpretation, which we now mention, is that this motif allows ZDHHC20 to interact with an ER protein, keeping it there. But of course, confirmation of this hypothesis would allow the identification of this protein. Alternatively, the PERW motif could prevent ZDHHC20 from getting out, hindering the interaction with proteins involved in export.

As a note for the reviewer, we know that the sequence is not strict. The mouse ZDHHC20^{Long} indeed contains a TESW sequence (extended data Fig 2a) instead of the PERW motif and still localizes to the ER (data not shown). We interrogated the human proteome (Ref seq annotated) for proteins harbouring a PERW or PERW-like motif and found more than 100 proteins with a PExW motif.

4. It seems that the explanation in the figure legend of Figure 4 j and l has changed.

We thank the referee for noticing this mistake. Figure 4 ordering has been corrected.

5. Graphic summary in last figure is strongly recommended to include.

Thank you, this indeed always help. We have included a Graphical model in Figure 5

Reviewers' Comments:

Reviewer #3:

Remarks to the Author:

The authors have addressed all my comments, expanding the discussion and clarifying some points as requested.

Reviewer #4:

Remarks to the Author:

The reviewer thinks that the points made in the paper have been adequately addressed. Thank you.